# Impacts of CICE sea ice model and ERA atmosphere on an Antarctic MetROMS ocean model, MetROMS-UHel-v1.0

Cecilia Äijälä[1], Yafei Nie[2], Lucía Gutiérrez-Loza[3], Chiara De Falco[3,6], Siv Kari Lauvset[3], Bin Cheng[4], David A. Bailey[5], and Petteri Uotila[1]

[1]University of Helsinki, Institute for Atmospheric and Earth System Research / Physics, Helsinki, Finland
[2]School of Atmospheric Sciences, Sun Yat-Sen University, and Southern Marine Science and Engineering Guangdong Laboratory (Zhuhai), Zhuhai, China
[3]NORCE Norwegian Research Centre, Bjerknes Centre for Climate Research, Norway
[4]Finnish Meteorological Institute. Polar meteorology and climatology research group, Helsinki, Finland
[5]NSF National Center for Atmospheric Research, Boulder, Colorado, USA
[6]Barcelona Supercomputing Center (BSC), Spain

**Correspondence:** Cecilia Äijälä (cecilia.aijala@helsinki.fi)

**Abstract.**

In recent years, the Antarctic sea ice has experienced major changes, which are neither well understood nor adequately reproduced by earth system models. To support model development with an aim to improve Antarctic sea ice and upper ocean predictions, the impacts of updating the sea ice model and the atmospheric forcing are investigated. In the new MetROMS-UHel-v1.0 (henceforth MetROMS-UHel) ocean-sea ice model, the sea ice component has been updated from CICE5 to CICE6, and the forcing has been updated from ERA-Interim to ERA5 reanalyses. The two versions of MetROMS evaluated in this study use a version of the regional ROMS ocean model including ice shelf cavities. We find that the update of CICE and ERA reduced the negative bias of the sea ice area in summer. However, the sea ice volume decreases after the CICE update but increases when the atmospheric forcing is updated. As a net result after both updates, the modelled sea ice becomes thinner and more deformed, particularly near the coast. The ROMS ocean model usually yielded a deeper ocean mixed layer compared to observations. Using ERA5, the situation was slightly improved. The update from CICE5 to CICE6 resulted in a fresher coastal ocean due to a smaller salt flux from sea ice to the ocean. In the ice shelf cavities, the modelled melt rates are generally underestimated compared with observations, with the largest underestimation coming from the ice shelves in the too cold Amundsen and Bellingshausen Seas as well as from the Australian sector in East Antarctica. These identified sea ice and oceanic changes vary seasonally and regionally. By determining sea ice and oceanic changes after the model and forcing updates, and evaluating them against observations, this study informs modellers on improvements and aspects requiring attention with potential model adjustments.

## 1   Introduction

The Southern Ocean is a key component in the global climate system, influencing the global ocean circulation and climate. In particular, sea ice plays a crucial role by regulating heat exchanges between ocean and atmosphere, controlling salt fluxes and

freshwater distribution, and significantly impacting the uptake and storage of $CO_2$ (Rintoul, 2018). In recent years, this region has experienced significant changes. Following a decades-long gradual overall increase, Antarctic sea ice is experiencing a rapid decrease after reaching a record high extent in 2014 (Parkinson, 2019). The minimum summer sea ice extent in 2022, 2023, and 2024 are the three lowest measured in 46 years of satellite records (Purich and Doddridge, 2023; Gilbert and Holmes, 2024; NSIDC, 2024). Furthermore, the winter sea ice maximum of 2023 was a record low (Purich and Doddridge, 2023; Gilbert and Holmes, 2024). Studies have shown an increased variance in Antarctic sea ice cover and a changing response to atmospheric forcing (Hobbs et al., 2024). Furthermore, the sea ice decline has been linked to ocean warming, which has been identified as the underlying cause of shifts in the sea ice regime (Purich and Doddridge, 2023).

The physical processes behind the recent observed changes are not yet fully understood, and further research is needed. However, studying a dynamic and remote environment, such as the Southern Ocean, is challenging and expensive. In situ observations in the Southern Ocean are sparse and time series short, especially for subsurface ocean measurements and in the sea ice zone, and they are biased towards summer conditions (Charrassin et al., 2008). Most common measurement systems face technical and logistical challenges in the Antarctic ice-covered areas. Surface moorings are hard to deploy due to the high possibility of ice related damage, and floats cannot always resurface to transmit data reliably (Rintoul et al., 2014). Satellite sensors have become indispensable in climate system observations, however, they cannot observe below sea ice and the persistent cloud cover limits their utility at high latitudes. Despite increasing efforts, the Antarctic Ocean below the sea ice and ice shelves remains among the least observed systems (Rintoul et al., 2014). And while satellite-based estimates of sea ice properties beyond ice concentration, such as sea ice thickness, are becoming available in the Arctic, these efforts are not nearly as well advanced in the Antarctic. Given the sparsity and complexity of the observations, accurate modelling of this area is particularly important.

Modelling Antarctic sea ice is also challenging. The northern boundary of the Antarctic sea ice is exposed to the Southern Ocean, which is a highly dynamic environment that is subjected to strong winds, waves and currents. The sea ice properties can vary greatly with ice floes of varying thickness, broken by cracks and leads, and usually covered in snow (Worby et al., 2008). Antarctic sea ice extends to lower latitudes if compared to its Arctic counterpart, and it has a stronger seasonality, while its thickness is limited by the relatively warm Circumpolar Deep Waters (CDW) (Maksym et al., 2015). The Southern Ocean receives more snowfall than anywhere else in the world (Lawrence et al., 2024), and the snow distribution on the sea ice is more complex than in the Arctic due to warmer winter temperatures and frequent snow flooding (Willatt et al., 2010). Model development and tuning have historically been focused on Arctic sea ice, and such efforts are not entirely transferable to Antarctica, making the modeling of Antarctic sea ice more challenging. Therefore, understanding how the different processes affecting the Antarctic sea ice work in models, and what shortcomings they have, is critical. Model development is continuously ongoing, and new versions of the different components need to be implemented in the models that use them. With better models, we could reduce model uncertainty and potentially improve future projections of the sea ice and the ocean.

This paper presents the circumpolar Antarctic coupled ocean–sea ice model MetROMS-UHel, an updated version of the coupled model, MetROMS-Iceshelf. MetROMS-Iceshelf has been used for studies on ice shelves, the effect of increased basal melt rate on the Antarctic Slope Current, and the transport of sea ice at the coast (Naughten et al., 2018b; Huneke et al., 2023).

However, model development and advances in reanalysis products providing higher resolution and near present forcing are both expected to have important implications in the model outputs (Barthélemy et al., 2018). Here, a new implementation of the model, including updates in the sea ice component and atmospheric forcing, is presented. Four simulations with different atmospheric forcing and sea ice module combinations are used to evaluate the impact of such updates on sea ice variables, ice shelf characteristics, and oceanic components. An overview of the model is presented in Sect. 2, while the experimental design is described in Sect. 3. The results are presented and discussed in Sect. 4.

## 2  Model description

MetROMS-UHel is an updated version of the coupled ocean/sea ice/ice shelf model MetROMS-Iceshelf (Naughten et al., 2017, 2018b). In this new version, the sea ice component has been updated with a newer version (see Sect. 2.1). Furthermore, the atmospheric forcing ERA-Interim (ERAI) (Dee et al., 2011) has been substituted with the latest ECMWF (European Centre for Medium-Range Weather Forecasts) reanalysis version, ERA5 (Hersbach et al., 2017).

### 2.1  Model overview

MetROMS-Iceshelf and MetROMS-UHel consist of the free-surface, terrain-following Regional Ocean Modelling System (ROMS) (Shchepetkin and McWilliams, 2005) and the dynamic-thermodynamic sea ice model CICE (Community Ice CodE) (Hunke et al., 2015, 2022) coupled with the Model Coupling Toolkit (MCT 2.11) (Larson et al., 2005; Jacob et al., 2005). The ocean physical component for both model versions, MetROMS-Iceshelf and MetROMS-UHel, is the same. They use the development Version 3.7 of the Rutgers ROMS code with ice shelf thermodynamics (Galton-Fenzi et al., 2012). The ocean setup for both models has a baroclinic timestep of 5 min, with 30 barotropic timesteps for each baroclinic timestep.

The coupled MetROMS models use different versions of CICE, specifically, MetROMS-Iceshelf uses CICE 5.1.2 (Hunke et al., 2015) while MetROMS-UHel uses CICE 6.3.1 (Hunke et al., 2022). CICE has been completely reworked between versions 5 and 6, with major restructuring and refactoring of the code, updated physics parametrization and bug fixes. These changes were not considered 'climate changing' in standalone mode, and most changes affected non-default physics. The developer defines 'climate changing' as 'significant changes in sea-ice thickness, h, over a substantial fraction of the ice pack within a defined number of annual cycles' (Roberts et al., 2018). CICE6 has been shown to improve the results of Arctic sea ice compared to CICE5 and CICE4 in the standalone mode (Wang et al., 2020b). However, some code changes, such as the ones related to salinity and fresh water flux calculations, might affect ocean coupled simulations. The model setup and parameters used for CICE in both models have been kept as similar as possible, using mostly CICE default values. Both versions use the elastic-viscous-plastic dynamics (Hunke and Dukowicz, 2003; Bouillon et al., 2013), which is the default dynamics option in CICE6. For advection, the incremental remapping scheme for sea ice by Lipscomb and Hunke (2004) is employed. For ice strength, the energetics-based approach of Rothrock (1975) is used, with ridging participation and redistribution functions by Lipscomb et al. (2007). Thermodynamics are handled by a mushy-layer thermodynamics scheme (Turner et al., 2013) with a level-pond parametrization (Hunke et al., 2013) with Stefan refreezing. We use the Delta–Eddington radiation scheme

(Briegleb et al., 2007; Holland et al., 2012), which calculates apparent optical properties, like albedo, from inherent optical properties. Both models use seven vertical ice layers, one snow layer and five ice thickness categories.

The original coupling was implemented by the Norwegian Meteorological Institute (Debernard et al., 2017), and described in Naughten et al. (2017). The updated coupling of CICE6 in MetROMS-UHel was implemented as part of this work, following the same principles of the original coupling. ROMS and CICE run on separate processors and communicate through the MCT coupler. The models have the same resolution (see Sect. 2.2), but ROMS uses an Arakawa C-grid while CICE uses an Arakawa B-grid (Arakawa and Lamb, 1977). Even though the position of the variables differs between these grids, both model components have the tracers at the cell centre, which is used for the coupling, with variables interpolated there as needed. The variables passed from CICE to ROMS are: sea ice concentration, ice–ocean stress vectors, heat, salt and freshwater fluxes, and shortwave radiation coming through the ice. The variables passed from ROMS to CICE are: sea surface temperature, sea surface salinity, ocean velocity averaged over the top 5 m, sea surface height, and the freeze–melt potential, which is the energy flux (in W m$^{-2}$) associated with the temperature difference from the freezing point. The freeze–melt potential, following (Naughten et al., 2017), is integrated over the top 5 m of the ocean model. The coupling timestep for both models is 30 min, the same as the thermodynamic timestep for CICE.

## 2.2 Domain

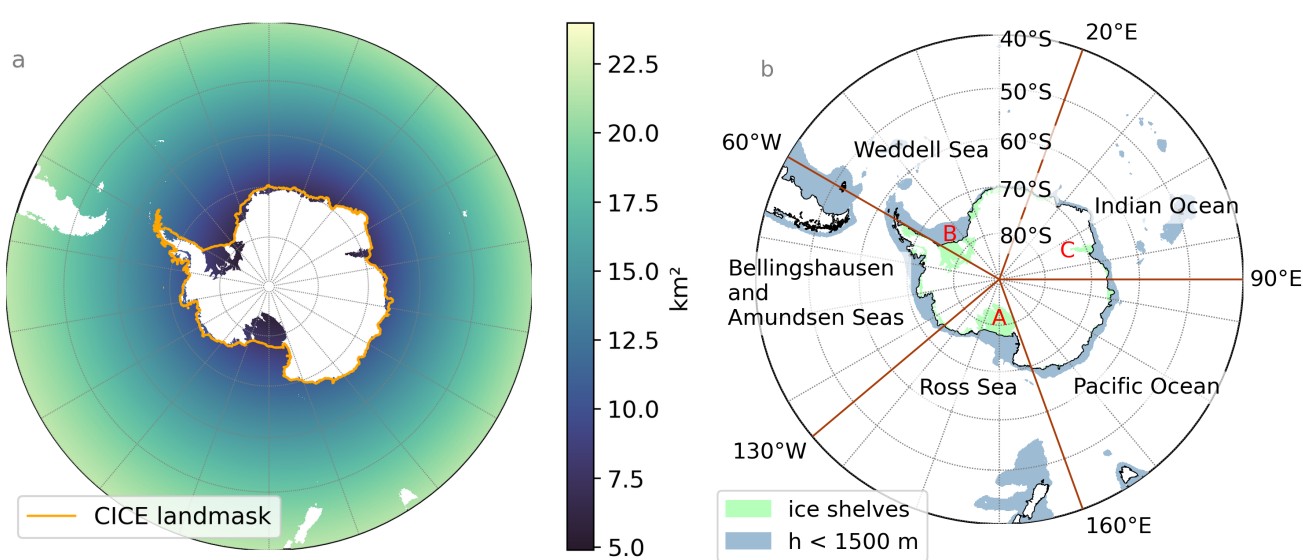

**Figure 1.** (a) Horizontal resolution of the domain in km$^2$ (from $40°$ S). The white area shows the ROMS land mask, and the orange line shows the edge of the CICE land mask around Antarctica. The area between these land masks contains the ice shelves that are included in ROMS, but not in CICE. (b) Map of the five Antarctic sectors used in the analysis. The light green shading shows the ice shelf cavities and the blue indicates areas with depths up to 1500 m, marking the continental shelves. The legends A, B and C in red indicate the three largest ice shelves: the Ross, Filchner-Ronne and Amery Ice Shelves, respectively.

MetROMS-UHel and MetROMS-Iceshelf use the same grid as in Naughten et al. (2017, 2018b). The grid is a circumpolar Antarctic quarter-degree grid scaled by cosine of latitude and with a relocated South Pole. The northern boundary is at 30° S, and in the south the model extends into the ice shelf cavities. The ice shelf draft is static and the bathymetry, ice shelf draft and land/sea mask are calculated from the RTopo-1.05 dataset (Timmermann et al., 2010) using a three-step smoothing procedure described in Naughten et al. (2018b) to ensure stability at the terrain-following coordinates.

The resulting Cartesian resolution of the grid cells is 8–10 km on the shelf and 15–20 km at the Antarctic Circumpolar Current (ACC, around 50° S). At the southernmost grounding lines of the Filchner-Ronne, Amery, and Ross Ice Shelves the grid resolution is 5 km or even finer (Fig. 1). The ocean model has 31 terrain-following vertical levels. This sigma coordinate results in a varying vertical resolution, being 1–3 m at the surface, and 200–300 m in the deep interior ocean. In the ice shelf cavities, the vertical resolution can often be less than 1 m.

## 3  Experiment design

**Table 1.** Description of the four ocean–sea ice model simulations analysed in this study.

| Experiment name | Model | CICE version | Atmospheric forcing | Modelling time |
|---|---|---|---|---|
| C5EI | MetROMS-Iceshelf | CICE 5.1.2 | ERAI | January 1992–August 2019 |
| C5E5 | MetROMS-Iceshelf | CICE 5.1.2 | ERA5 | January 1992–April 2023 |
| C6EI | MetROMS-UHel | CICE 6.3.1 | ERAI | January 1992–August 2019 |
| C6E5 | MetROMS-UHel | CICE 6.3.1 | ERA5 | January 1992–April 2023 |

Four experiments with different atmospheric forcing and sea ice module combinations are used (see Table 1) to evaluate the performance of the new MetROMS-UHel in comparison to MetROMS-Iceshelf. These simulations allow us to assess the impacts of the updated sea ice model and those attributed to the atmospheric forcing on the model outputs. Two of the simulations, C5EI and C5E5, use the MetROMS-Iceshelf setup where the old sea ice component CICE5 is used, but each simulation is forced with a different atmospheric reanalysis (ERAI and ERA5, respectively). Similarly, the simulations C6EI and C6E5 using the MetROMS-UHel setup with the updated sea ice component CICE6 are forced with ERAI and ERA5, respectively. All simulations start in January 1992 and run until the end of the forcing data (August 2019 for ERAI and April 2023 for ERA5). The initial conditions, spinup, and forcing are described in the two following sections.

### 3.1  Initial conditions and spinup

The ROMS ocean initial conditions for temperature and salinity are taken from the ECCO2 Cube 92 reanalyses (Menemenlis et al., 2008; Wunsch et al., 2009), using data from January 1992. Temperature and salinity have been extrapolated to the ice shelf cavities using the nearest neighbour method. Initial velocities and sea surface height are set to 0.

The initial conditions for the sea ice in CICE are created using an ice mask which sets ice where the NOAA/NSIDC Climate Data Record of Passive Microwave Sea Ice Concentration Version 2 (Meier et al., 2013) shows concentrations greater than 15 %. In these areas, the model initializes the ice to be 1 m thick with 100 % concentration and 0.2 m of snow on top of the sea ice.

All experiments are initialised with a spinup, when the model is run for 5 years with the forcing for year 1992 before switching to yearly forcing. The spinup is not long enough for the ocean model to reach an equilibrium state. However, the spinup is long enough for the sea ice and upper ocean to reach equilibrium and for the total kinetic energy of the regional model to even out. Therefore, for the analysis of the surface and continental shelves, we argue that a longer spinup is not required. The basal melt rates of the ice shelves take until year 1996 to even out (8 years) and basal melt is therefore analyzed for the period 1996–2018 (Sect. 4.3). The deep ocean does not reach equilibrium and some model drift can be seen in the interior ocean, for example, in the ACC transport, measured at the Drake Passage (Appendix A)which decreases. A similar, but less negative trend of the ACC was found in the MetROMS-Iceshelf runs by Naughten et al. (2018b).

## 3.2 Forcing

We run the models with atmospheric forcing from two different reanalyses from ECMWF, ERAI and the newer ERA5. ERA5 is a considerable upgrade from ERAI with a finer horizontal resolution of 31 km compared to the ERAI 80 km. The underlying IFC (Integrated Forecasting System) modelling system has seen multiple improvements over ten years, with development in model physics, numerics, and data assimilation (Hersbach and de Rosnay, 2018). The atmospheric reanalyses are interpolated to the model grid, and the ERA5 hourly data are interpolated to the same temporal resolution that is used for ERAI. We use 6-hourly near-surface temperature, pressure, humidity, winds, and total cloud cover. Additionally, we use 12-hourly rain, snow, and evaporation. The 6-hourly data are linearly interpolated in the model for each timestep (5 min for ROMS and 30 min for CICE), and the 12-hourly data are applied at a constant rate with a step change every 12 h. The largest differences between the ERA5 and ERAI forcing (Fig. B1) are observed mainly at the coast, where the increased resolution helps resolve small-scale processes, for example for the wind, that see large changes at the coast. Freshwater flux from iceberg melt is also added using a monthly averaged 100-year climatology from Martin and Adcroft (2010) as in Naughten et al. (2018b). The model does not have tidal forcing.

The same salinity restoration scheme is applied to the surface of ROMS in all the model runs, as previously used in MetROMS-Iceshelf by Naughten et al. (2018b), where it was used to prevent deep convection in the Weddell Sea. Salinity from the World Ocean Atlas 2013 (WOA13) monthly climatology by Zweng et al. (2013) is linearly interpolated for each model timestep and repeated annually. The salinity restoration is applied on a 30-day timescale to the uppermost layer. Following Naughten et al. (2018b), the restoration is not applied on the Antarctic continental shelf (i.e., areas south of 60° S where the depth is less than 1500 m) nor in the ice shelf cavities (see Fig. 1b) to reduce the excessive freshening of Antarctic Bottom Water (AABW).

For the northern boundaries in ROMS, we use monthly ECCO2 reanalysis temperature, salinity, and horizontal velocity, while zonal velocity is clamped to 0, such that there is flow through the border but not along it. The numerical methods used

to apply boundary conditions are the Radiation-Nudging scheme (Marchesiello et al., 2001) for baroclinic meridional velocity, temperature, and salinity. This scheme combines a radiation condition with nudging to external data. The radiation condition, combined with a weak nudging, minimizes the reflection of internal information propagating out of the domain and avoids drifting. A stronger nudging is applied to incoming data. The Flather scheme (Flather, 1976) is used for barotropic meridional velocity, where differences in modelled and boundary sea surface height are used to tune the normal velocity. This method avoids over-specification due to simple clamping (Mason et al., 2010) but still allows mass-balance conservation (Carter and Merrifield, 2007). The Chapman scheme (Chapman, 1985) for sea surface height, which is often used in combination with the Flather scheme, is derived from the linearized gravity wave equation and designed to minimize the reflection of model gravity waves propagating outwards (Mason et al., 2010). For sea surface height, we use the AVISO climatology (AVISO, 2011). To make the transition into the domain, we apply a constant topography in the $y$ direction to the 15 northernmost cells, which corresponds to around $3°$ in latitude. CICE does not require northern boundary conditions, as the sea ice edge does not come close to the northern boundary of the model.

## 4  Results and Discussion

### 4.1  Sea Ice

#### 4.1.1  Sea Ice Concentration and Area

The modelled sea ice concentration is compared to satellite observations from NOAA/NSIDC Climate Data Record of Passive Microwave Sea Ice Concentration, Version 4 (Meier et al., 2021), referred to as NSIDC CDR hereafter. The ice edge is defined as the point where sea ice concentration reaches 15 %. It's important to note that sea ice concentrations below this threshold are uncertain in satellite measurements (Zwally et al., 1983), which should be considered when interpreting the results. In general, all simulations underestimate the sea ice concentration compared to NSIDC CDR (Fig. 2), but updating the model and atmospheric forcing generally increases the concentration (Fig. B2). However, some large-scale patterns are observed, including both underestimation and overestimation of the sea ice concentration, that are consistent throughout all model runs.

During the summer (JFM), the largest underestimations of the sea ice concentration are observed in the western region of the Weddell Sea sector and close to the coast and ice edge in the Indian Ocean, Pacific Ocean, and the Bellingshausen and Amundsen Seas (BellAm hereafter) sectors (Fig. 2e, i, m, q). Sea ice concentration is overestimated in the eastern Weddell Sea and the Ross Sea sectors, primarily north of the ice edge (black line in Fig. 2), except for C6E5, where the majority of the Ross Sea sector is also overestimated (Fig. 2q). Updating the sea ice model from CICE5 to CICE6 decreases the sea ice concentration close to the coast, especially in the Indian Ocean sector (Fig. B2i), while both the change from ERAI to ERA5 and from CICE5 to CICE6 increase the concentration in the Ross Sea, BellAm, and Weddell Sea sectors (Fig. B2m), which decreases the underestimation compared to the satellite measurements seen in these sectors south of the ice edge.

In autumn (AMJ), the sea ice concentration is mostly underestimated in the models compared to NSIDC CDR satellite measurements (Fig. 2f, j, n, r), but not as much as in summer. However, closer to the coast where sea ice concentrations are

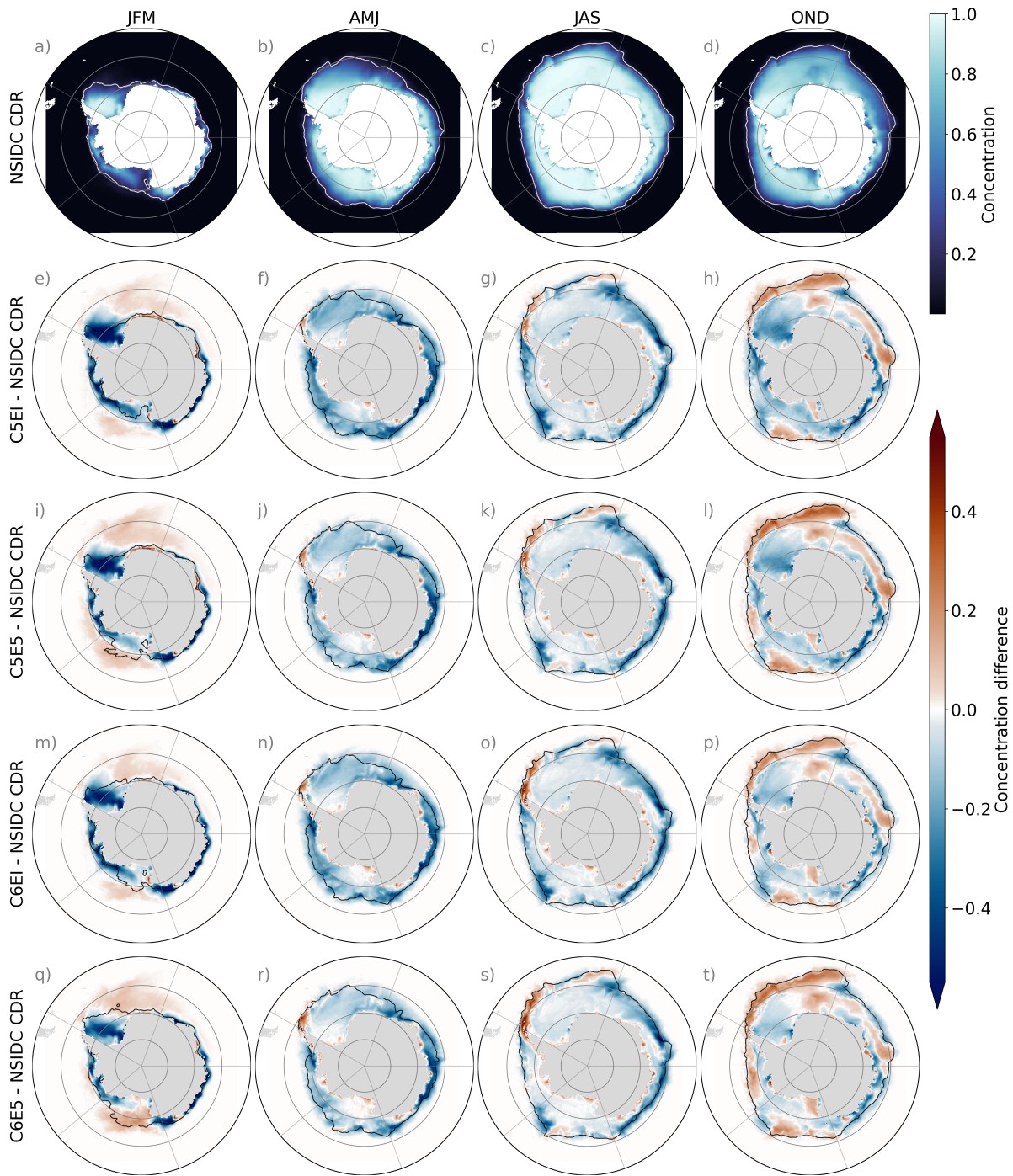

**Figure 2.** (a–d) Seasonal mean sea ice concentration over the period 1992–2018 from NSIDC CDR and (e–t) concentration differences between NSIDC CDR and the different model runs (C5EI, C5E5, C6EI, C6E5). The white (black) line shows the sea ice edge of the observations (model). The sea ice edge is defined using a 15 % sea ice concentration threshold.

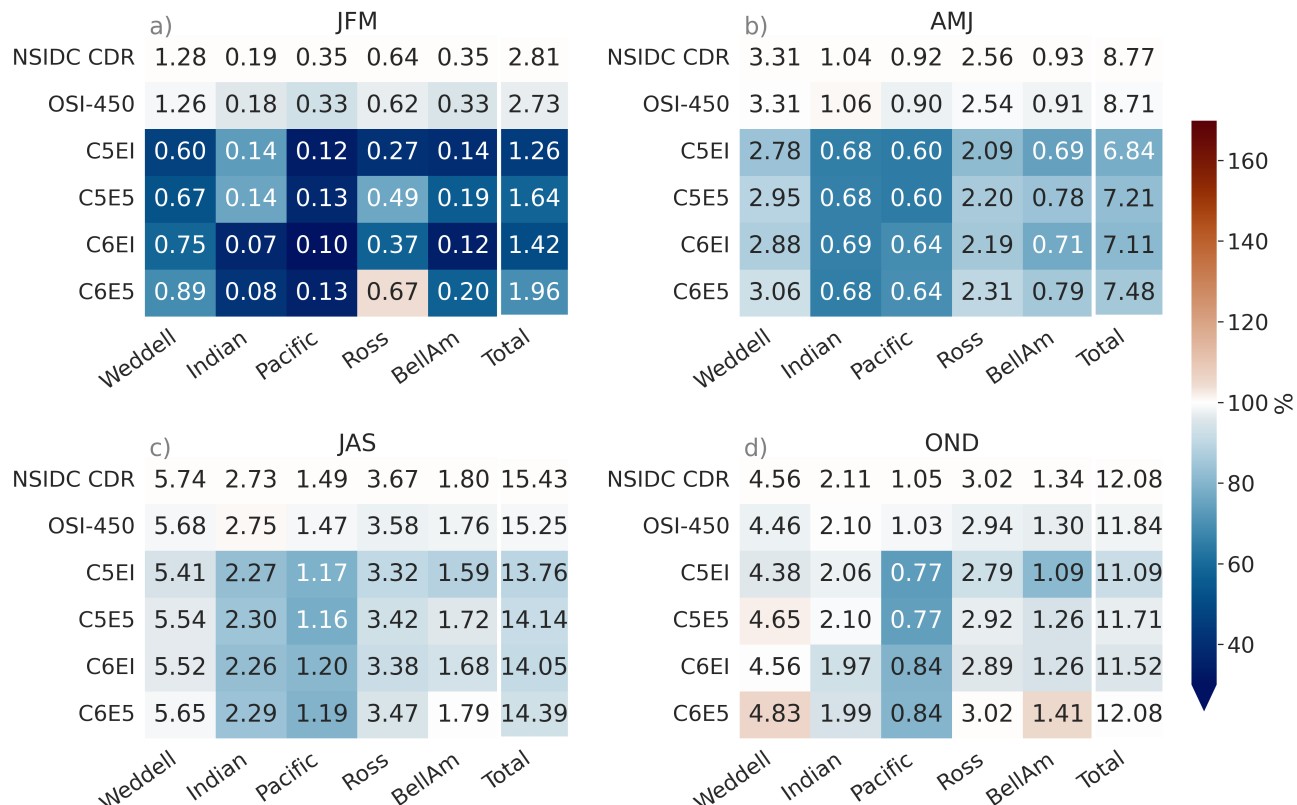

**Figure 3.** Seasonal means of sea ice area for the period 1992–2018 in the different Antarctic sectors (Fig. 1b). The first two rows are satellite observations from NSIDC CDR and OSI-450. The last four rows are the different model runs (C5EI, C5E5, C6EI, C6E5), The numbers are the area of sea ice [$10^6$ km$^2$], and the colour is the fraction of the area compared to the NSIDC CDR *([model area]/[NSIDC CDR area])*.

close to 100 %, the models perform well, particularly south of 70° S in the Weddell Sea and Ross Sea sectors. The models overestimate the sea ice close to the ice edge in the Antarctic Peninsula at the border of the BellAm and Weddell Sea sectors and in some coastal areas.

The differences between the modelled and the observed sea ice concentration during winter (JAS) show similar spatial patterns to those observed during the autumn (AMJ) (Fig. 2g, k, o, s). However, along the ice edge, an increased overestimation is observed in the Weddell Sea sector, while large underestimations occur in the other sectors. The differences in sea ice concentration among the different model runs are largest close to the ice edge, especially in the BellAm and Weddell Sea sectors, where the largest concentrations can be found from C6E5 (Fig. B2o).

The spatial pattern of the differences in the concentration between the model runs and NSIDC CDR is more variable in spring (OND) than in the other seasons, with both underestimation and overestimation happening in the entire domain (Fig. 2h, l, p, t). Nevertheless, the observed patterns are consistent throughout the model runs. The overestimation happens mostly close

to the ice edge and large parts of the Indian Ocean sector, while underestimation occurs largely in coastal areas and the Weddell and Ross Seas. On the one hand, changing the atmospheric forcing from ERAI to ERA5 tends to increase the concentration in the Weddell Sea, Ross Sea, and BellAm sectors (Fig. B2h). On the other hand, updating the sea ice model from CICE5 to CICE6 decreases the concentration close to the ice edge in the Weddell Sea, Indian Ocean, and Ross Sea sectors, and at most of the coastline, but increases the concentration otherwise (Fig. B2l).

The modelled sea ice area is compared to NSIDC CDR and the Global Sea Ice Concentration Climate Data Record, OSI-450 (OSI SAF, 2022). The area is calculated from the concentration for both the satellite products and the model simulations as the sum of the sea ice concentration (>15 %) multiplied by the cell area. The four model runs have, generally, a smaller total area compared to the satellite observations (Fig. 3). Overall, the sea ice area from C6E5 is the closest to the NSIDC CDR satellite measurements during all seasons, while C5EI has the largest differences. The C5E5 run is the second closest to the observations, which implies that the change from ERAI to ERA5 has generally a stronger effect than the change from CICE5 to CICE6. When comparing the satellite products, OSI-450 has a slightly lower total area than NSIDC CDR. Nevertheless, the OSI-450 total area is still consistently larger than the model runs in all seasons, except spring (OND). Throughout the year, the largest underestimations happen in the Pacific and Indian Ocean sectors, where especially in the Pacific sector the shelf water seems to be too warm (Fig. B6), and the warm CDW seems to get too far south (not shown).

During the summer (JFM), every model run underestimates the area compared to NSIDC CDR and OSI-450 in all sectors, except for C6E5 in the Ross Sea (Fig. 3a). Underestimated summer sea ice is a common bias in both standalone ocean models and coupled models (Naughten et al., 2018b; Goosse et al., 2023; Roach et al., 2020; Schroeter and Sandery, 2022). Furthermore, differences between the modelled and the observed total sea ice area are largest during the summer (JFM), as is the variability between the different model runs. Differences in the modelled sea ice areas can be seen especially in the Weddell Sea and Ross Sea sectors where C6E5 is closest to the satellite measurements, and in the Indian Ocean sector where the ice-covered area drops by half when changing from CICE5 to CICE6, placing C6EI and C6E5 further away from the satellite reference values.

Through autumn (AMJ) and winter (JAS), the differences between satellite and modelled sea ice areas are smaller than in the other seasons. Overall, model runs have smaller sea ice areas than the satellite observations, with C6E5 being the closest to the satellite measurements in all sectors (Fig. 3b, c).

During spring (OND) models show more variability with both overestimation and underestimation of the sea ice area compared to the observations in the different sectors and model runs (Fig. 3d). The Pacific sector consistently shows underestimations in all model runs during spring (OND), while clear overestimations are observed in the Weddell Sea in the models forced with ERA5 (i.e. C5E5 and C6E5), as well as in the BellAm sector from the C6E5 simulation. Nevertheless, the overall differences between modelled sea ice areas and NSDIC CDR are the smallest during this season as underestimations and overestimations cancel each other out (see Fig. 2), except in the Pacific sector.

Additionally, the temporal changes of the minimum and maximum sea ice area throughout the period 1992–2022 are analysed. The model outputs from February (minimum) and September (maximum) sea ice area are compared to NSIDC CDR and the Sea Ice Index, Version 3 (Fetterer et al., 2021) (Fig. 4). The NSIDC CDR sea ice concentration is calculated as the

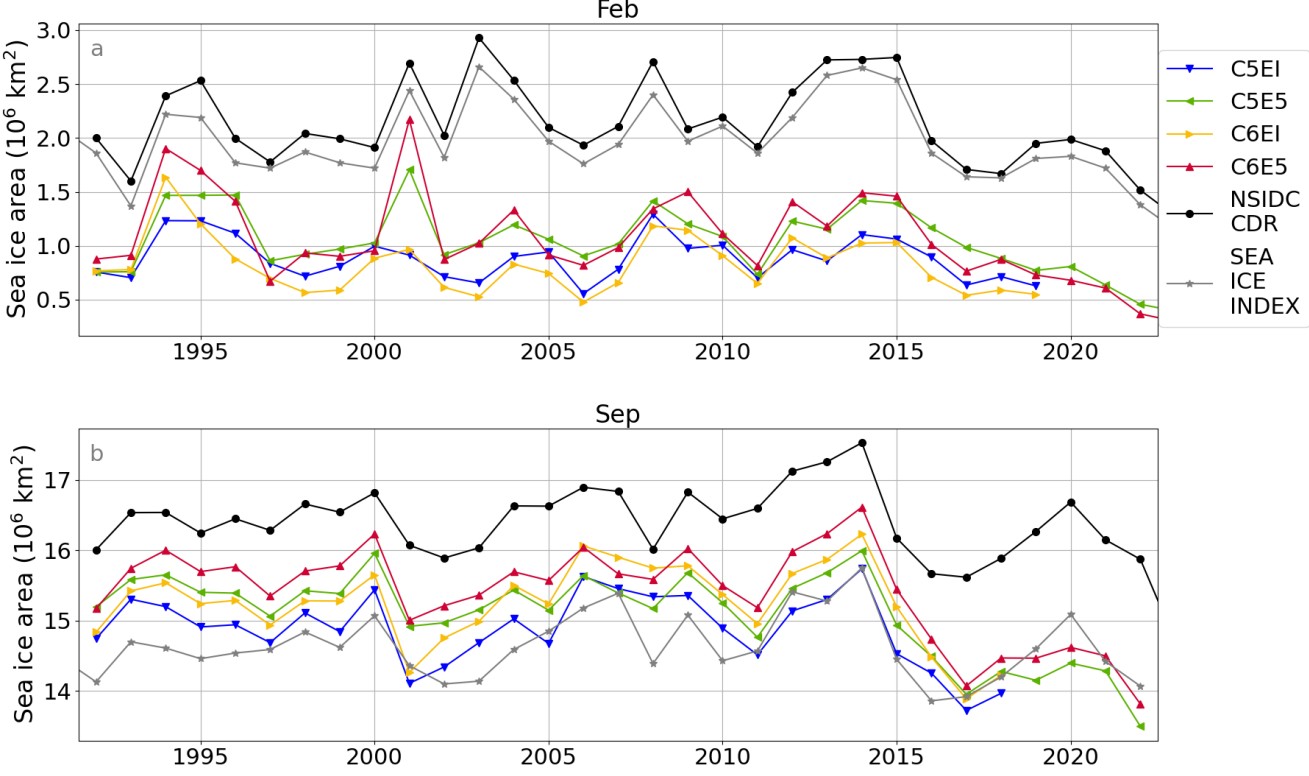

**Figure 4.** Timeseries over 1992–2022 of (a) February and (b) September monthly average sea ice area [$10^6$ km$^2$] for the four different model runs (blue, green, yellow, red) and Sea Ice Index (grey), as well as NSIDC CDR sea ice concentration (black). Further information about the means and linear trends for the periods 1992–2014 and 2014–2018, as well as the correlation with NSIDC for 1992–2018, can be found in table B1.

maximum of the NASA Bootstrap (Comiso and Nishio, 2008) and the NASA Team algorithms (Cavalieri et al., 1997), while the Sea Ice Index uses only the NASA Team algorithm. From these, the NASA Team algorithm tends to get lower concentration

values in the pack ice than the bootstrap method (Screen, 2011), and thus has a lower area. In both months, all the model runs follow the observed year-to-year variation relatively well (Fig. 4). For example, the decline in the sea ice area around 2015 and its recovery until 2020, with a decline again in 2021 and 2022 can be seen, even though the degree of recovery is smaller in the models than in the observations. Overall, the correlation to NSIDC (Table B1) is mostly affected by the atmospheric forcing update in February, with the largest correlation in simulation C5E5 at 0.68 and lowest in C6EI at 0.47. Meanwhile, the sea ice

model update has a stronger effect in September, with the strongest correlation in C6E5 at 0.87 and weakest in C5EI at 0.80.

In February, both the Sea Ice Index and the NSIDC CDR datasets show larger sea ice areas than the model runs, with the Sea Ice Index being slightly lower than NSIDC CDR (Fig. 4a), with 1992–2014 means of 2.05 $\times 10^6$ km$^2$ and 2.23 $\times 10^6$ km$^2$, respectively (Table B1). The simulations forced with ERA5 (C5E5 and C6E5) show, most of the time, larger values of

minimum sea ice area than those forced with ERAI (C5EI and C6EI), with C6E5 having the largest mean in 1992–2014 at
1.18 $\times 10^6$ km$^2$ and C6EI the smallest at 0.86 $\times 10^6$ km$^2$ (Table B1). Such a pattern is consistent with the seasonal sea ice
area means in Fig. 3. The differences between experiments using CICE5 and CICE6 are not systematic and vary from year to
year. In September, modelled sea ice areas are between the Sea Ice Index (1992–2014 mean 14.74 $\times 10^6$ km$^2$) and the NSIDC
CDR (1992–2014 mean 16.56 $\times 10^6$ km$^2$) (Fig. 4b, Table B1). Out of the four model runs, the C6E5 run shows the highest sea
ice area (1992–2014 mean 15.70 $\times 10^6$ km$^2$), closer to NSIDC CDR, while C5EI gives lower values (1992–2014 mean 15.02
$\times 10^6$ km$^2$) similar to those from the Sea Ice Index.

Overall, the results for concentration and area are consistent with each other. Updates in the atmospheric forcing from ERAI
to ERA5, and the sea ice model from CICE5 to CICE6, are shown to reduce the underestimation of both sea ice concentration
and area, relative to satellite observations (e.g., NSIDC CDR). The change in atmospheric forcing has, on average, a larger
effect on the sea ice area than the sea ice model. The underestimation of the sea ice is largest in the summer (JFM). For models
using ERAI forcing, low summer sea ice has been attributed to a bias of low cloud cover in ERAI (Naud et al., 2014), which
leads to excessive sea ice melt (Naughten et al., 2018b). ERA5 has also been shown to exhibit a similar bias (Wang et al.,
2020a), although the biases in cloud cover are smaller in ERA5 than in ERAI (Wu et al., 2023). Additionally, the atmospheric
forcing has been shown to have a warm bias in surface temperature in certain regions. For example, in the Weddell Sea,
both atmospheric reanalyses show a warm surface bias compared to buoy measurements, though this bias is smaller in ERA5
(King et al., 2022). Spatial differences within and between sectors are noticeable and not always consistent throughout the
simulations, so they cannot be directly attributed to the changes in atmospheric forcing or the sea ice model.

### 4.1.2 Sea ice edge

In addition to sea ice area and concentration, we look at the ice edge in the model and compare it to satellite measurements.
For this, we use the integrated ice edge error (IIEE), defined as the area where the model and the observations disagree on the
ice concentration being above or below 15 %. The IIEE is calculated following Goessling et al. (2016), and using NSIDC CDR
satellite observations as comparison:

$$\text{IIEE} = O + U \tag{1}$$

where

$$O = \int_A \max(c_m - c_o, 0) \, dA \tag{2}$$

and

$$U = \int_A \max(c_o - c_m, 0) \, dA \tag{3}$$

where $A$ is the area of interest, here the Southern Ocean, $c$=1 if the sea ice concentration is >15% and $c$=0 otherwise, and $m$
and $o$ denote the model and the observations.

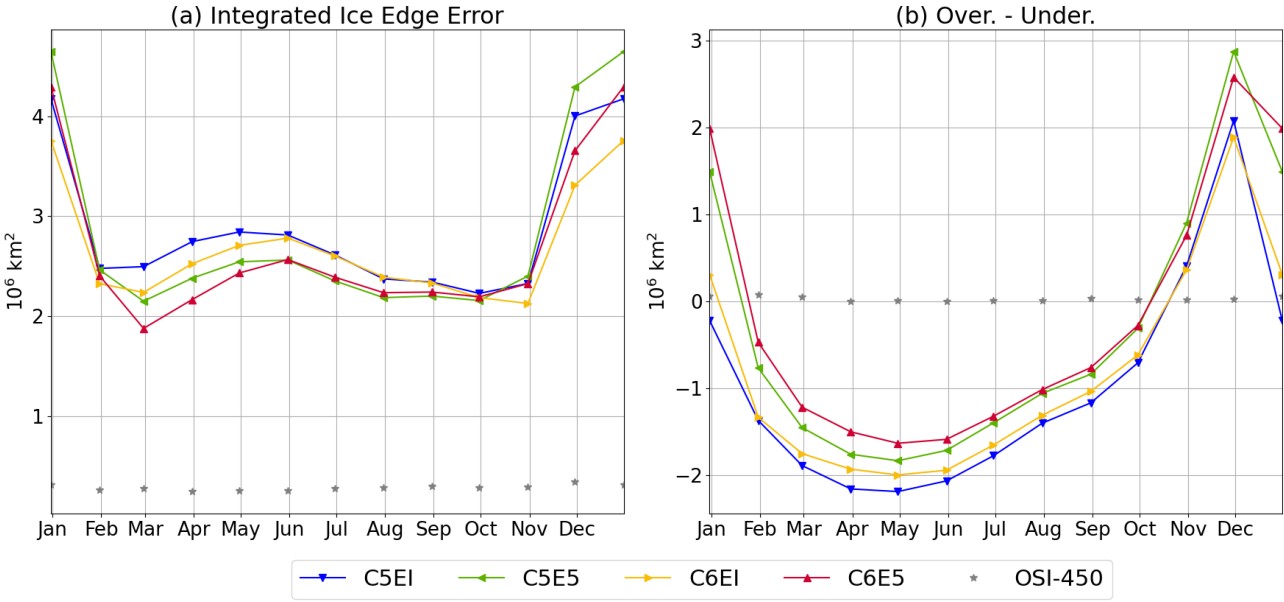

**Figure 5.** (a) Integrated ice edge error (IIEE) and (b) integrated ice edge overestimation-underestimation for all models compared to NSIDC CDR. Ice edge error for OSI-450 satellite measurements (grey stars) is presented to illustrate observational uncertainty. The analysis period is 1992–2018.

The smallest IIEE values are found between February–November for all model runs, with values ranging between 1.88–2.84×$10^6$ km$^2$. The largest IIEE is found from November to February, with maximum values in January (3.76–4.65×$10^6$ km$^2$). Simulations using ERA5 (C5E5 and C6E5) have a smaller IIEE (1.88–2.57×$10^6$ km$^2$) than the ERAI runs (2.23–2.84×$10^6$ km$^2$) from February to October. The CICE6 runs (C6EI and C6E5) have smaller values from February to June than the CICE5 runs (C5EI and C5E5), while from June to October both CICE versions have similar errors. From October to February, the C6EI run has the smallest error (2.13–3.76×$10^6$ km$^2$) and the C5E5 run the largest (2.16–4.65×$10^6$ km$^2$).

The difference between the ice edge overestimation minus the underestimation (Fig. 5b) indicates the direction of the IIEE. Positive values of this difference indicate a net overestimation of the ice edge in the model, suggesting an average ice edge located northward of the observed ice edge. Negative values indicate a net underestimation in the model, with the modelled ice edge located, on average, to the south of the observed ice edge. Further, values closer to zero indicate a good model performance in comparison to the observations in terms of the net ice edge. From February to October, the four model runs underestimate the ice edge (Fig. 5b). These results are consistent with the sea ice concentration and area underestimations found in summer (JFM), autumn (AMJ), and winter (JAS) in all model runs (see Fig. 2 and 3). These results indicate that the modelled ice edge from all simulations is located southward from the observed ice edge. As discussed in Sect. 4.1.1, the atmospheric forcing seems to play a larger role than the sea ice model, in this case, reducing the underestimation of the sea

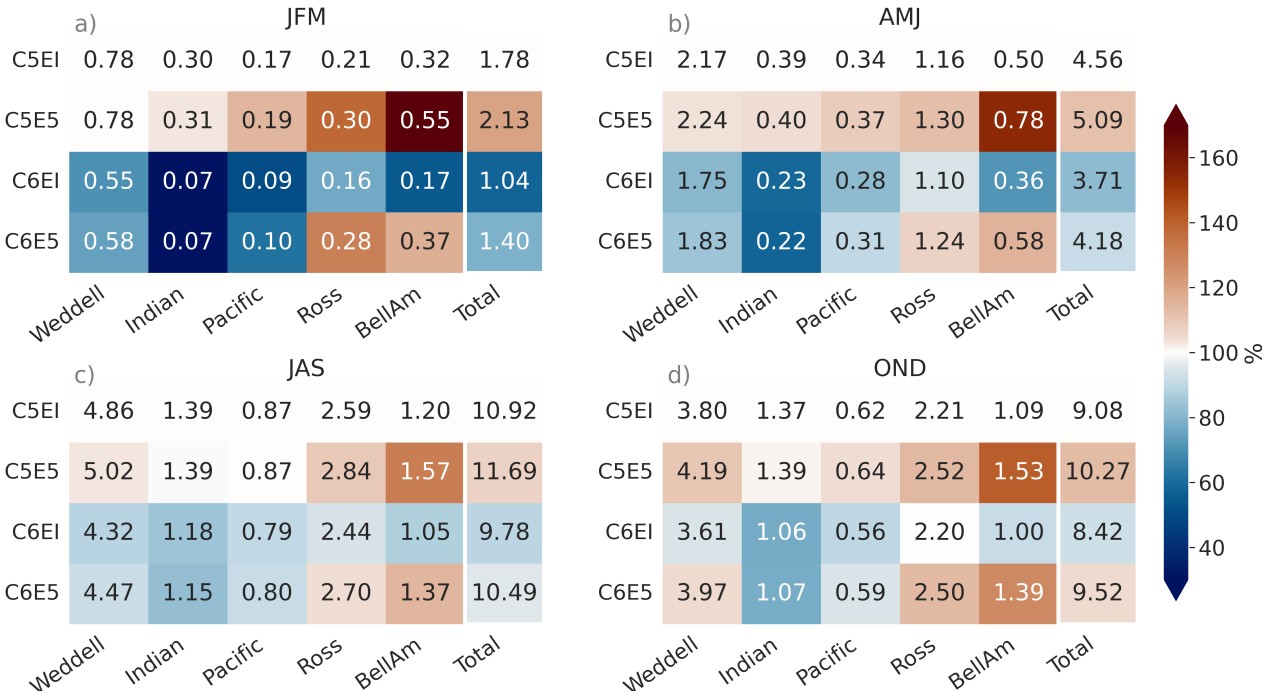

**Figure 6.** Seasonal mean sea ice volume [$10^3$ km$^3$] over 1992–2018 for the different sectors shown in Fig. 1b. The colour stands for the fraction of the area compared to the C5EI model run *([model volume]/[C5EI volume])*.

ice edge. Simulations forced with ERA5 (C5E5 and C6E5) result in a larger average sea ice extent than ERAI runs (C5EI and
C6EI), better representing the ice edge conditions from the satellite observations between February–October. On the contrary, from November to January when an ice edge overestimation is observed from all model runs, the update in atmospheric forcing from ERAI to ERA5 increases the overestimation. The update from CICE5 to CICE6 results in smaller IIEE year-round for the simulations forced with ERAI, i.e. smaller underestimations between February–October and smaller overestimations in November–January (December for C5EI). The sea ice model update in the ERA5 simulations results in an IIEE decrease,
particularly between February–June, otherwise resulting in small improvements throughout the year.

### 4.1.3  Sea ice volume

Sea ice thickness observations in the Southern Ocean are scarce and have large uncertainties (Holland et al., 2014; Uotila et al., 2019; Xu et al., 2021), largely due to the complexity in the snow cover, which makes it challenging to measure freeboard of the sea ice from satellites and affecting satellite-based estimates of ice thickness (Giles et al., 2008). For this reason, the sea
ice volume results are assessed based solely on the comparison between the four different simulations using experiment C5EI as a baseline.

The total sea ice volume decreases due to the sea ice model update from CICE5 to CICE6, while the change in atmospheric forcing from ERAI to ERA5 has an opposite effect, resulting in the C5E5 simulation having the largest and C6EI the smallest volume in all seasons (Fig. 6). This is different from how the sea ice area behaves as both the sea ice model update and the atmospheric forcing update increase the area (see Sec. 4.1.1). The summer (JFM) is the season with the highest variability between simulations (Fig. 6a), as was the case for the area and concentration.

The increase in volume caused by the update of the atmospheric forcing to ERA5 is most pronounced in the BellAm and Ross Sea sectors, where all seasons show an increase in volume in both C5E5 and C6E5 (Fig. 6). In particular, C5E5 exhibits a significant volume increase in the BellAm sector during summer (JFM) and autumn (AMJ) where there is a systematic increase in volume in the whole sector (Fig. B3e–h). The large increase in volume is especially noticeable at the Antarctic Peninsula, in both the BellAm and Weddell Sea sectors. However, in the Weddell Sea, the sea ice volume is smaller in C5E5 than in the C5EI baseline along the ice shelf edge, resulting in only a moderate net overestimation in the sector (Fig. 6 and Fig. B3e–h). The volume increase in C5E5 compared to C5EI is mostly consistent with the increase in sea ice concentration (Sect. 4.1.1, Fig. B2e–h). However, the low sea ice volume observed in the coastal region of the Weddell Sea is not followed by a decrease in concentration, indicating thinner ice.

The volume decrease caused by the sea ice model update is largest in the Indian Ocean sector, where a clear decrease in concentration is also seen (Sec. 4.1.1). Simulation C6EI consistently shows a smaller sea ice volume than the C5EI baseline in all sectors around the year (Fig. 6). The largest negative differences are observed along the coast across all sectors (Fig. B3i–l). Nevertheless, the sea ice volume is larger than in the baseline further away from the coast during spring (OND), resulting in a smaller net negative difference. In this simulation (C6EI), the volume and concentration patterns are not consistent, and only the summer (JFM) shows a decrease in both volume and concentration along the coast (Fig. B2i–l).

In C6E5, where both the atmospheric forcing and the sea ice model are updated, the difference from C5EI falls between C5E5 and C6EI. The total sea ice volume is less than in C5EI in all seasons except spring (OND) (Fig. 6), indicating that the update from CICE5 to CICE6 has a greater impact than the update of the atmospheric forcing. However, since C6E5 has a larger volume than C5EI in the Ross Sea and BellAm sectors throughout the year, it suggests that the atmospheric forcing has a stronger influence in these sectors. Similar to C5E5 and C6EI, the C6E5 simulation shows a smaller sea ice volume than C5EI close to the coast while further away from the coast it shows larger volumes than C5EI (Fig. B3m–p).

The increased concentration (see Sect. 4.1.1) but decreased volume when updating from CICE5 to CICE6, as seen in the difference between C6EI and C5EI (Fig. B3i–l and B2i–l), can be potentially caused by a difference in level ice area fraction (alvl) and ridged ice area fraction (ardg). C6EI has a larger alvl and smaller ardg than C5EI (Fig. B4). This means that the CICE6 model runs have less thick ridged ice and more thin level ice than the CICE5 model runs. The distribution of the difference in ardg (Fig. B4e–h) between the C6EI and C5EI shows similar patterns as the volume difference between the model runs, with the strongest decrease at the coast. Meanwhile, the difference in the distribution of alvl (Fig. B4a–d) is similar to the difference in concentration, except around the coast where there is an increase in alvl from C5EI to C6EI in all seasons other than summer (JFM), that does not occur in the ice concentration. Ridging has been shown to increase sea ice volume (Zhang,

2014). Therefore, less ridging in C6EI is consistent with the smaller sea ice volume observed, while the large volume in C5EI, especially during winter, results from the larger ice growth in the open water regions caused by the ridging.

### 4.1.4 Sea ice growth and melt

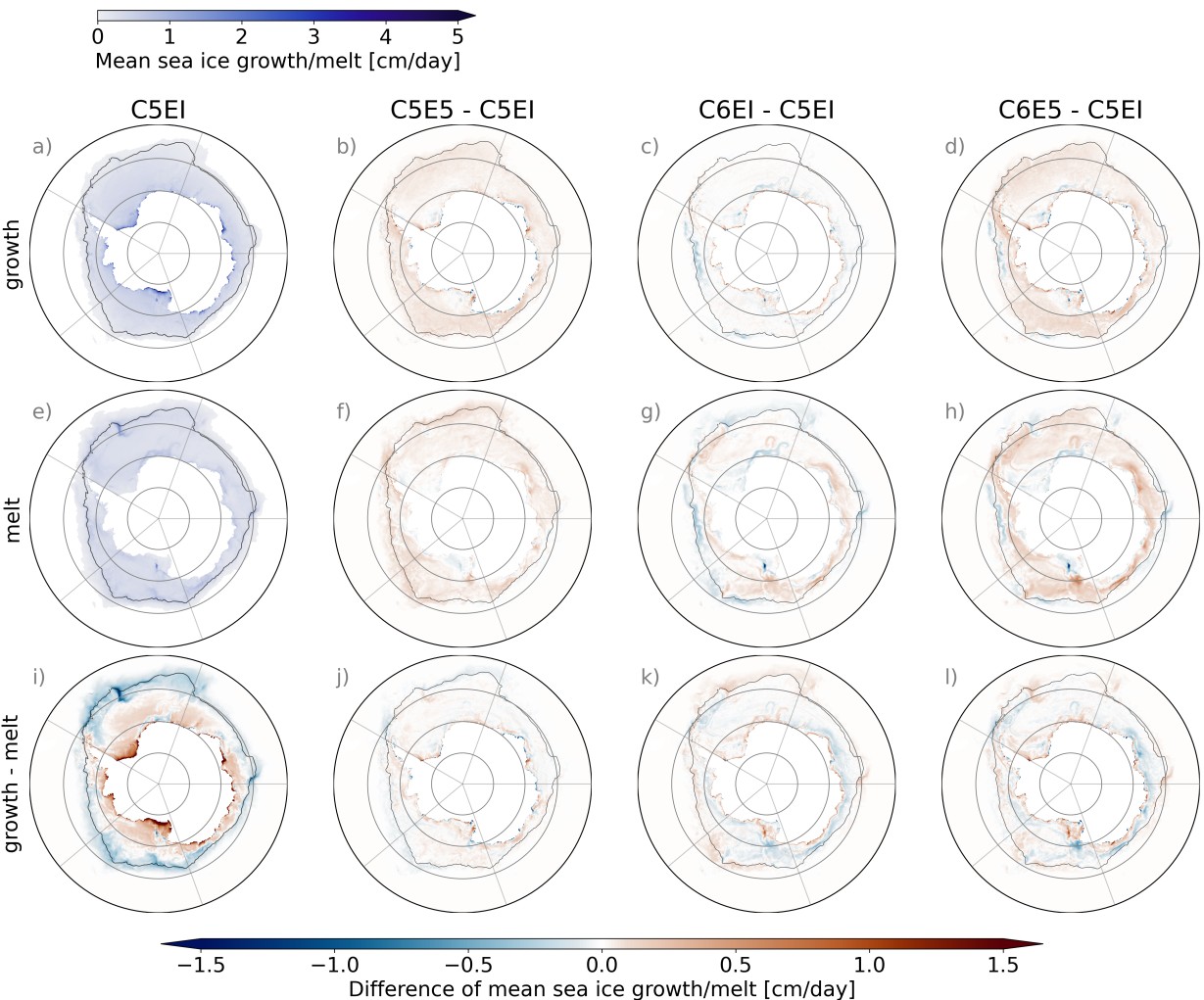

**Figure 7.** Mean sea ice growth (frazil ice growth+congelation ice growth+snow ice formation) and melt (top ice melt + basal ice melt + lateral ice melt) for C5EI (a, e) and difference between the other models and C5EI (b–c, f–h) during 1992–2018. The last row (i–l) shows the difference between the growth and melt above, for C5EI (i) and the model differences (j–k).

The average sea ice growth was calculated from congelation ice growth, frazil ice growth and snow ice formation, of which congelation ice growth is the largest. The strongest sea ice growth occurs along the coast (Fig. 7a), where strong winds, blowing

from land, create polynyas and areas with lower sea ice concentration. The ERA5 sea ice growth clearly increases compared to ERAI (Fig. 7b), primarily due to the snow ice growth increase, which is the result of increased ERA5 snowfall compared to ERAI (Fig. B1h). This is in line with the sea ice volume growth discussed in Sect. 4.1.3. Compared to ERAI, the ERA5 sea ice growth increases everywhere, except right at the coast, where it on average decreases, although with fine-scale regional variations. The most pronounced decreases are observed in the Pacific and Indian sectors, whereas the largest increases occur at the edges of the three major ice shelves, Filchner-Ronne, Ross, and Amery, as well as in the eastern Weddell sector and at the tip of the West Antarctic Peninsula, in the eastern BellAm sector. Regarding decadal all-season average ERA5 and ERAI differences, there is no clear association with these differences in ice growth and wind (Fig. B1e,f). However, regarding seasonal differences, the decrease in sea ice growth from ERAI to ERA5 aligns quite well with the corresponding reduction in wind speed, especially in autumn (AMJ) and winter (JAS) (not shown).

The transition from CICE5 to CICE6 (Fig. 7c) does not substantially change the total amount of sea ice growth. The sea ice production increases along most of the coast, but reduces further away. The most notable exceptions, i.e. decreases in growth at the coast, occur on both sides of the Filchner-Ronne Ice Shelf, west of the Amery Ice Shelf, and in front of most of the Ross Ice Shelf, particularly on its western side. The most notable areas of reduction in sea ice growth occur further offshore at parts of the ice edge in the BellAm and Ross sectors, just north of the continental shelf in the eastern Weddell sector, and to a small but consistent extent in the Indian and Pacific sectors. The sea ice growth and concentration increase, while the sea ice volume decreases close to the coast from CICE5 to CICE6. This supports the argument, presented in Sect. 4.1.3, that the decrease in volume is connected to the change in ridged and level ice area fractions (Fig. B4). The combined effect of these changes, as implemented in C6E5 (Fig. 7d), results in, on average, increased sea ice growth. Compared to C5EI, the ice growth increases in most of the offshore regions, except at the ice edge in the BellAm sector, and in parts of the Weddell and Ross Seas. Along the coast, both increased and decreased growth are observed, though a larger part of the coastline experiences an increase rather than a decrease.

Unlike the growth, sea ice melt is more evenly distributed (Fig. 7e). The latitude where melt is consistently larger than growth depends on the sector, being further south of 70° S in the middle of the BellAm sector and almost as north as 60° S in the eastern Weddell Sea sector in C5EI (Fig. 7i). Similar to growth, melt rates are generally higher in the ERA5 simulations (Fig. 7f), largely due to increased bottom melt (not shown). Nevertheless, the ERA5 melt rates are lower than ERAI close to the coast and over the continental shelf, especially in spring and summer (OND, JFM). While no clear correlation due to the atmospheric forcing change is evident from multidecadal all-season averages, the reduction in melt at the coast in spring (OND) and summer (JFM) coincides with a slightly lower ERA5 air temperature for those seasons (not shown). Furthermore, a decrease of ERA5 melt at the ice edge coincides mostly with areas where its volume and area, or both, are smaller than in ERAI, especially in spring (OND) (Fig. B2l and B3l). Switching the sea ice model from CICE5 to CICE6 does not substantially change the total melt. It results in a decrease in melt close to the ice edge and the coast, with increases occurring in between (Fig. 7g). In CICE6, the decrease in growth and increase in melt in most of the Indian and Pacific sectors, except at the coast (Fig. 7k), is partially responsible for the decreased area and volume in those regions compared to CICE5 (Fig. 3 and 6). When

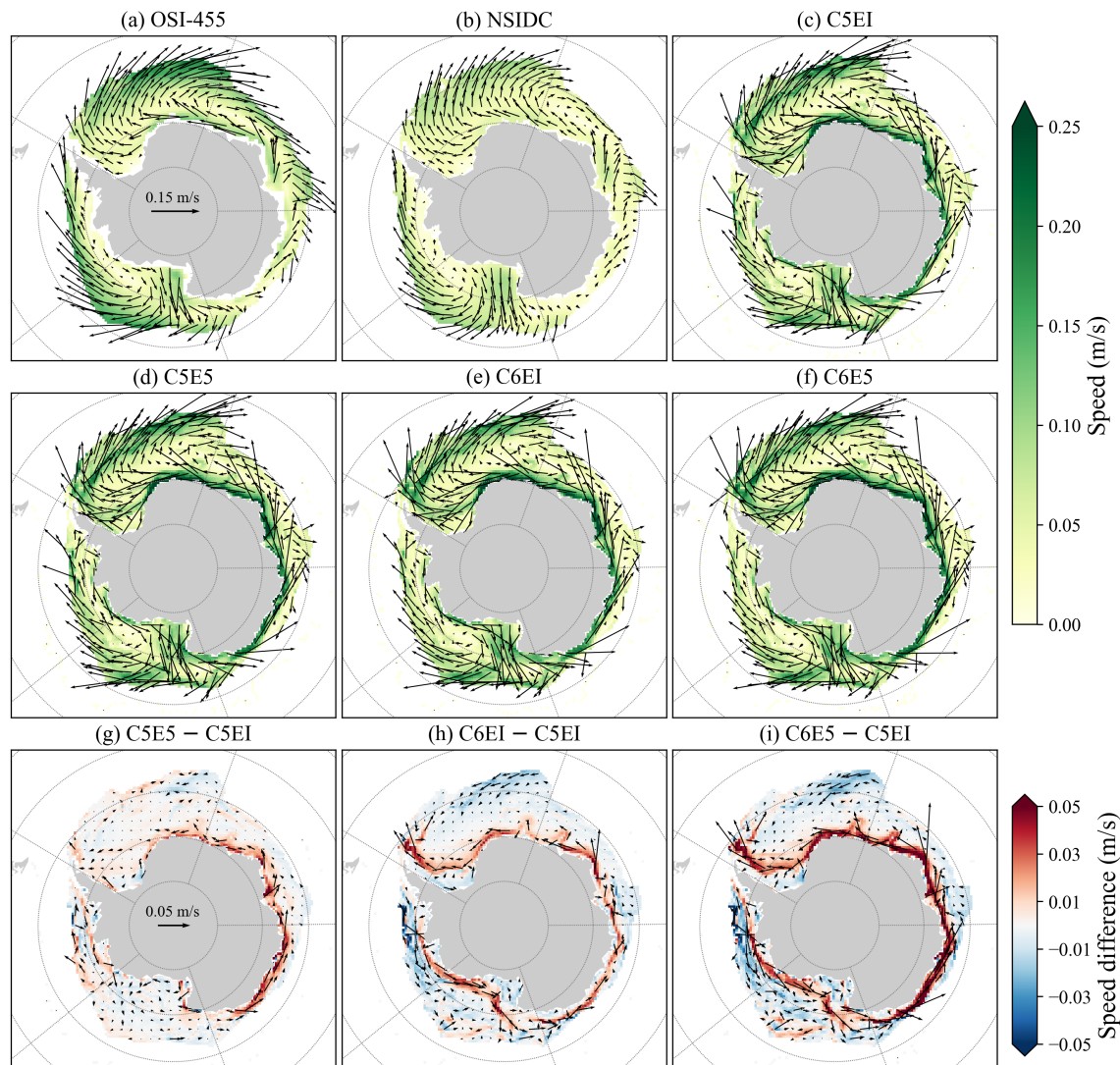

**Figure 8.** Mean sea ice velocities during winter (JAS) over the period 1992–2018, overlaid on the speed from (a) OSI-455, (b) NSIDC ice drift and (c–f) model outputs, and differences between sea ice velocities between model runs (g–i). Only grid cells with climatological winter (JAS) sea ice concentration larger than 15 % are plotted.

both changes are applied together, as implemented in C6E5 (Fig. 7h), the resembling pattern emerges but with bigger increase in melt compared to C5EI, as the forcing update from ERA5 to ERAI dominates.

### 4.1.5 Sea ice drift

The sea ice drift was compared to two observational ice velocity products: 1) the Global Low Resolution Sea Ice Drift data record Release 1 from EUMETSAT (European Organisation for the Exploitation of Meteorological Satellites), hereafter OSI-

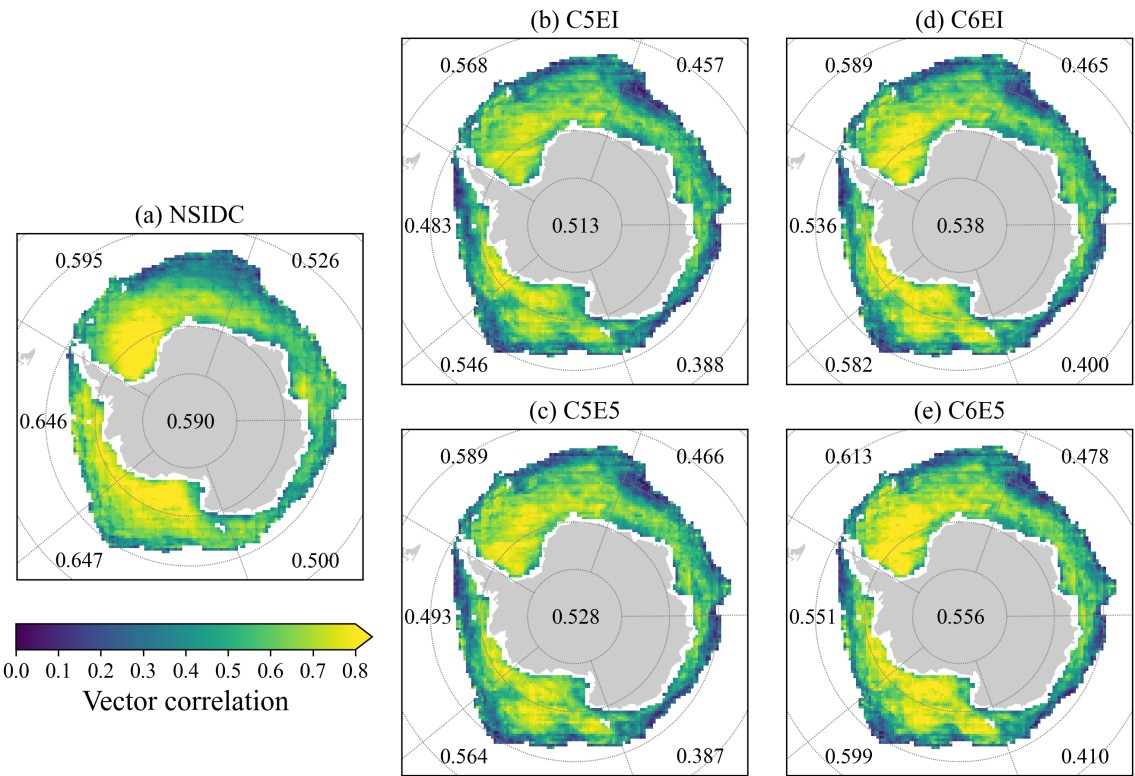

**Figure 9.** Significant ice drift vector correlation coefficients from monthly mean data during 1992–2018 at a confidence level of 99 % between OSI-455 data and (a) NSIDC ice drift and (b–e) model outputs. The mean correlation coefficients for each geographic sector are listed in their respective positions, with the value in the centre of the figure representing the pan-Antarctic mean.

455 (OSI SAF, 2022), with a spatial resolution of 75 km, and 2) the NOAA/NSIDC Polar Pathfinder Daily 25 km EASE-Grid Version 4, hereafter NSIDC ice drift (Tschudi et al., 2019). For NSIDC ice drift, the meridional velocity was multiplied with a constant factor of 1.357 to correct for its underestimated speed, as in Haumann et al. (2016).

All simulated and observed sea ice drift velocities are bilinearly interpolated onto a common polar stereographic grid with a resolution of 60 km to facilitate comparison. As satellite-derived ice velocities are often spatially smoothed to reduce the noise (Kimura et al., 2013), a 4 x 4 cell filter is first used to smooth the high resolution MetROMS-modelled ice velocities to ensure a comparable pattern with the observations.

Figure 8a–f shows the winter (JAS) climatological sea ice drift vectors and speeds, and 8g–i shows the difference between models. The two observational data sets are consistent in velocity direction, while NSIDC's ice speeds are consistently smaller than those of OSI-455. When comparing the model outputs with the observations, we should keep in mind that the coastal pixels are poorly resolved in the satellite data and are thus more uncertain.

As prescribed atmospheric forcing drives the model runs, the simulations generally captured the ice drift circulation patterns well (Fig. 8). For example, significant ice transport from the Ross Shelf and transitions from the westward coastal drift to the

eastward drift in the outer region, especially in the Ross Sea gyre and Weddell Sea gyre regions, can be seen. However, although the observations have considerable uncertainties near the Antarctic coast, clear overestimations can be seen. The observed large speeds could explain the biases seen in the sea ice concentration in spring (OND), discussed in Sect. 4.1.1. Furthermore, in the model, the larger speeds might occur due to the missing land-fast ice component. Furthermore, the relatively low resolution of the ocean-land boundary at the coast and the lack of grounded icebergs could also contribute to the overestimation. A higher resolution ocean-land boundary including icebergs would, potentially, cause slower average motion of the sea ice and longer surviving ice in summer (Naughten et al., 2018b). The low resolution ocean-land boundary might also be a reason for the underestimation of the sea ice, especially in the summer, as it has been shown that sea ice transport is an important process during melt season (Goosse et al., 2023). Updating from CICE5 to CICE6 or replacing ERAI forcing with ERA5 does not result in substantial changes, except for an increase of ice velocities near the coast in both updates, with the largest increase in the C6E5 run (Fig. 8). This is consistent with stronger coastal wind speeds (Fig. B1e,f) and reduced sea ice volume at the coast as discussed in Sect. 4.1.3.

To quantify their similarities, we also compute the vector correlation coefficients (VCCs) between NSIDC/modelled ice drifts and the OSI-455 using the Sea Ice Evaluation Tool (Lin et al., 2021) (Fig. 9). Both NSIDC and simulations show higher VCCs with OSI-455 in the western geographical sectors (Ross Sea, BellAm, and Weddell Sea) than in the east (Indian and Pacific Oceans). The correlation is particularly high in the areas of the Ross Sea gyre and the Weddell Sea gyre. Quantitatively, the improvements from replacing ERAI with ERA5 are now clearer than those shown in Fig. 8, with increases in VCCs for all sectors other than the Pacific Ocean. Updating the sea ice model from CICE5 to CICE6 gives a larger improvement than updating the atmospheric forcing, especially in the BellAm sector, the Ross Sea gyre, and the Weddell Sea gyre. The overall consistency between NSIDC and OSI-455 (0.590) is only slightly higher than between the simulations and OSI-455 (0.513–0.556), and not as pronounced as in the visualization of Fig. 8. C6E5 shows the highest correlation of the model runs, and C5EI is the lowest.

## 4.2 Oceanic variables

### 4.2.1 Ocean Hydrography

Oceanic heat content and basal melt are influenced by currents and water mass properties, which are, in turn, influenced by the sea ice and the atmosphere. In this section, we investigate the water mass properties on the continental shelf, south of 65° S (Fig. 10) and in the upper 1500 m over the entire Pan-Antarctic domain (Fig. 11). For this, we compare the model simulations to the 1° latitude×longitude gridded analysis of the observational dataset EN4.2.2 (Good et al., 2013) ensemble members using Gouretski and Cheng (2020) and Gouretski and Reseghetti (2010) bathythermograph corrections (later referred to as EN4). This dataset is maintained by the Met Office Hadley Centre for Climate Change. It provides comprehensive subsurface ocean temperature and salinity (vertical profiles and monthly objective analyses), integrating several data sources. It is important to keep in mind, when interpreting the results, that observational data from the Southern Ocean are sparse resulting in significant uncertainties in the EN4 data.

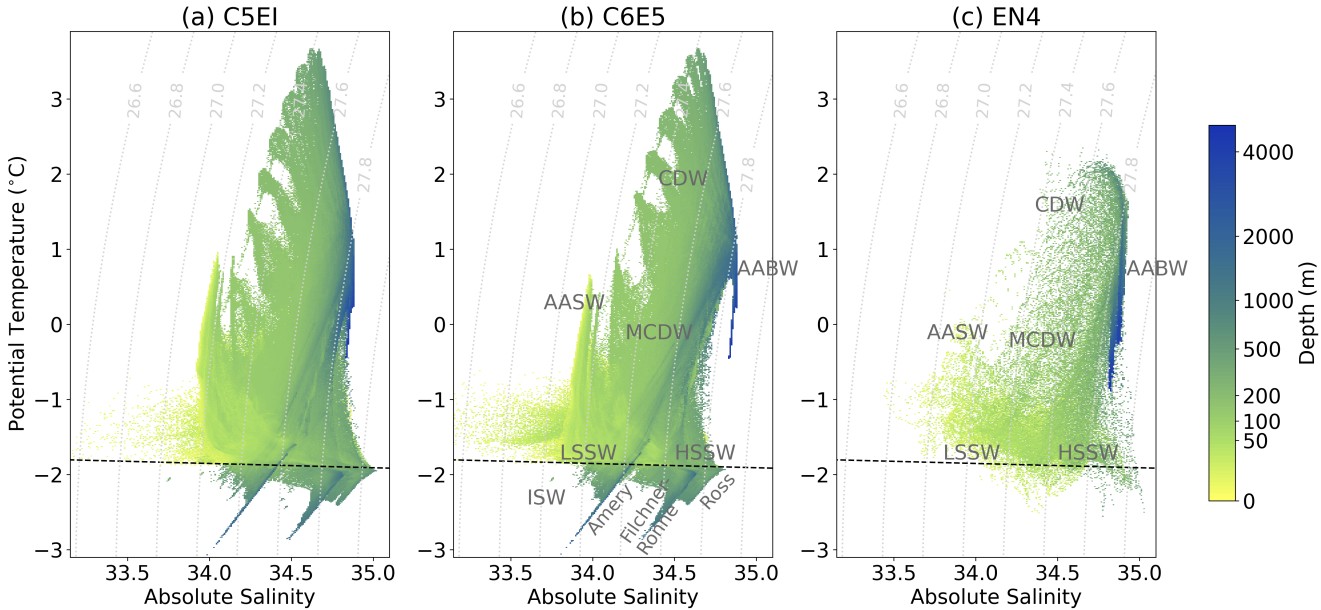

**Figure 10.** Modelled average 1992–2018 TS-diagrams for (a) C5EI, (b) C6E5 and (c) observed EN4 on the continental shelf (south of 65°
S). The absolute salinity is calculated from model output (psu) using the TEOS-10 standard. Depth indicates the average depth of a given
temperature–salinity value. The surface freezing point is marked with the dashed line. Water masses are marked in grey on the figure in
the middle: AABW: Antarctic Bottom Water, CDW: Circumpolar Deep Water, MCDW: Modified Circumpolar Deep Water, LSSW: Low-
Salinity Shelf Water, HSSW: High-Salinity Shelf Water, AASW: Antarctic Surface Water, and ISW: Ice Shelf Water.

On the continental shelf, all major water masses can be identified from the simulations (Fig. 10). Here, only model runs
C5EI and C6E5 are shown as the main differences come from the upgrade from CICE5 to CICE6, while changes from ERA5
to ERAI are relatively small (not shown). The overall shapes of the modelled TS-diagrams look similar. However, the Ice Shelf
Water (ISW), the High-Salinity Shelf Water (HSSW), the Low-Salinity Shelf Water (LSSW), and the Modified Circumpolar
Deep Water (MCDW) are consistently fresher in CICE6 compared to CICE5 runs (Fig. 10a, b).

Below the surface freezing point (dashed lines in Fig. 10) lies the ISW. This water mass can be this cold without freezing
because it is generated at the base of ice shelves, where the high pressure due to depth lowers the freezing point. The water
under the ice shelves follows the diagonal dilution ratio of melting and freezing seawater (Gade, 1979). The ice shelf waters
have different salinity, which can be used to differentiate them. The three biggest ice shelves in Antarctica are (in order of
decreasing salinity): the Ross, the Filchner-Ronne, and the Amery Ice Shelves (Fig. 1b). All of them are clearly fresher in the
C6E5 than in the C5EI run (Fig. 10).

A possible consequence of this salinity bias is that the freshening of coastal waters induced by the update to CICE6 disrupts
the export of dense water from the shelf, potentially shutting it down. Figure B5 presents the streamfunction values south of
60° S, but does not show any apparent differences between the evolution of CICE5 and CICE6 AABW cells. This might be

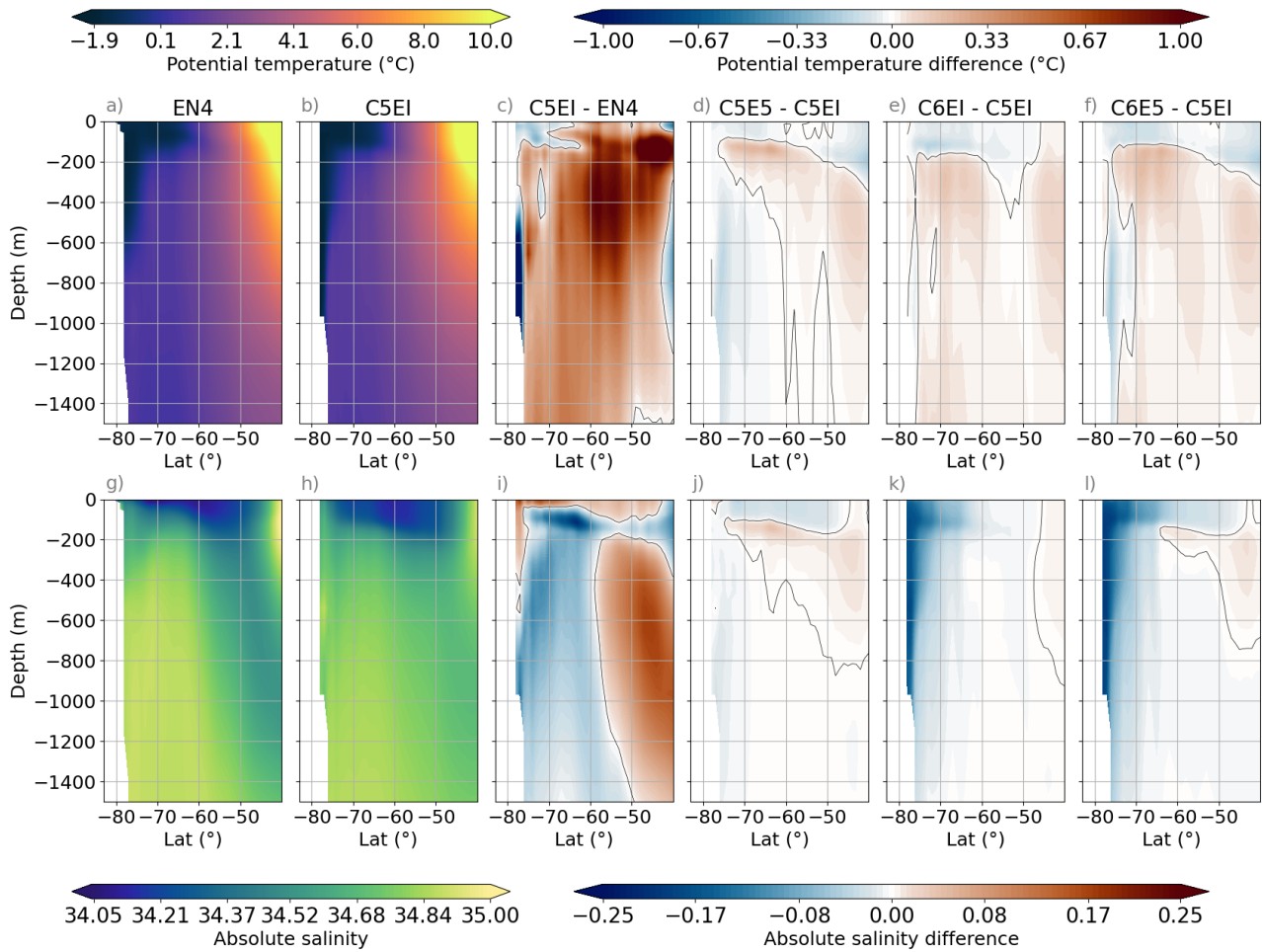

**Figure 11.** Pan-Antarctic zonal average of the 1992–2018 mean temperature and salinity from the upper 1500 m for EN4 observations (a, g), model run C5EI (b, h), difference between them (C5EI–EN4) (c, i) and model run differences C5E5-C5EI (d, j), C6EI-C5EI (e, k) and C6E5-C5EI (f, l). The grey contours in c–f (i–l) indicate a zero potential temperature (salinity) difference. The absolute salinity is calculated using TEOS-10 standard. Simulations have been interpolated to the EN4 grid, and ice shelf cavities have been left out of the plot, to make the data more comparable.

because the AABW cells are weak from the beginning and can not get any weaker in CICE6 despite the freshening of coastal waters.

The surface (above 100 m), which consists of Antarctic Surface Water (AASW), generally has salinity lower than 34.5 overlying the higher salinity subsurface waters. Salinity can be below 33.5 in the narrow embayments in the BellAm sector. Temperature is mostly below 0°C except in the BellAm sector. Surface waters that are more saline than 34.5 with temperatures 450  around -1°C can be found in the Ross Sea polynya. These waters are less saline in the CICE6 runs than in the CICE5 runs.

Just above the surface freezing point is the cold subsurface (~100–500 m) shelf water which is the result of sea ice formation. This water is divided into HSSW (salinity > 34.62: Miller et al. (2024)) and LSSW. Similar to the ISW, these water masses are fresher in C6E5 than in C5EI. Furthermore, CICE6 has a smaller salinity range between HSSW and LSSW, as the CICE6 to CICE5 HSSW salinity difference is larger than the CICE6 to CICE5 LSSW one. In general, these cold shelf bottom waters are less saline in the model than in EN4 (Fig. 10). Since EN4 data are quite sparse, we also compare the simulated shelf bottom salinity and temperature to the Schmidtko et al. (2014) climatology, compiled from seven publicly available CTD datasets (Fig. B6). This comparison confirms that the CICE6 model runs are fresher than the observations.

Deep waters take a long time to spinup due to longer residence times, as discussed in Sect. 3.1, and will therefore be the same or very similar to the initial conditions, and should be interpreted with caution. The deepest waters, below 1000 m, corresponding to the Antarctic Bottom Water (AABW), have salinity above 34.6 in all simulations (Fig. 10). The deepest, most saline parts of AABW are similar in all model runs, and are slightly warmer than in EN4, while the shallower parts of the AABW have a larger spread in salinity in the CICE6 simulations than the CICE5 simulations. Above the AABW, on the continental slope, the fresher and warmer CDW can be identified. The warmest CDW is located in the BellAm sector with modelled temperatures of up to 3°C, which are clearly warmer than in EN4 (Fig. 10). The model runs tend to be slightly warmer than EN4 in all the other sectors as well. The CDW mixes with other water masses and produces the cooler Modified CDW (MCDW), also noticeable in Fig. 10.

Pan-Antarctic zonal mean temperature and salinity are illustrated in Fig. 11. We focus our analysis on the top 1500 m as changes in the deeper ocean are small, and uncertain due to the short spinup. The temperature distribution is similar between EN4 and C5EI (Fig. 11a, b), but C5EI is warmer than EN4, except for the southernmost parts and above 200 m depth south of 60° S (Fig. 11c). Atmospheric forcing and sea ice model upgrades thus both contribute to a decrease of temperature in the upper ocean (Fig. 11d–f). Differences between C5E5 and C5EI also show a temperature increase below the mixed layer, with the maximum warming at ∼ 150 m in coastal areas (65° S), and a slight cooling below that and to the south (Fig. 11d). The upgrade to CICE6 leads to uniform warming of the ocean interior (Fig. 11e, f).

The vertical salinity gradient is weaker in the model simulations than in EN4 south of 60° S (Fig. 11g–i). The surface is more saline in the model run C5EI than in EN4 and less saline below the surface and south of 60° S with the largest negative difference in salinity at around 100 m depth between 60–70° S (Fig. 11i). Northward of 60° S, the relatively fresh Antarctic Intermediate Water (AAIW) is clearly visible in both EN4 and the models, but is much saltier in the models below 200 m depth. Updating the atmospheric forcing from ERAI to ERA5 slightly increases the gradient, freshening the surface slightly and increasing the salinity in the interior (Fig. 11j). The change of the sea ice model from CICE5 to CICE6 results in a clear freshening in the south (Fig. 11k), in line with what we could see on the shelves (Fig. 10). This freshening is also visible in the ice shelf cavities (not shown). Therefore, CICE6 runs have an increased fresh bias, with salinity values further away from the observations than CICE5 runs. This bias can be attributed to the salt flux from CICE to ROMS which is, on average, smaller in CICE6 than in CICE5. The largest differences can be found at the coast and when the salinity flux is positive, i.e. from the ice towards the ocean, where the ice growth is largest (Fig. 7). The problem seems to originate from a bug in the Icepack 1.3.1 code (submodule of CICE 6.3.1), regarding calculations of salinity fluxes from frazil ice formation (CICE-Consortium, 2022),

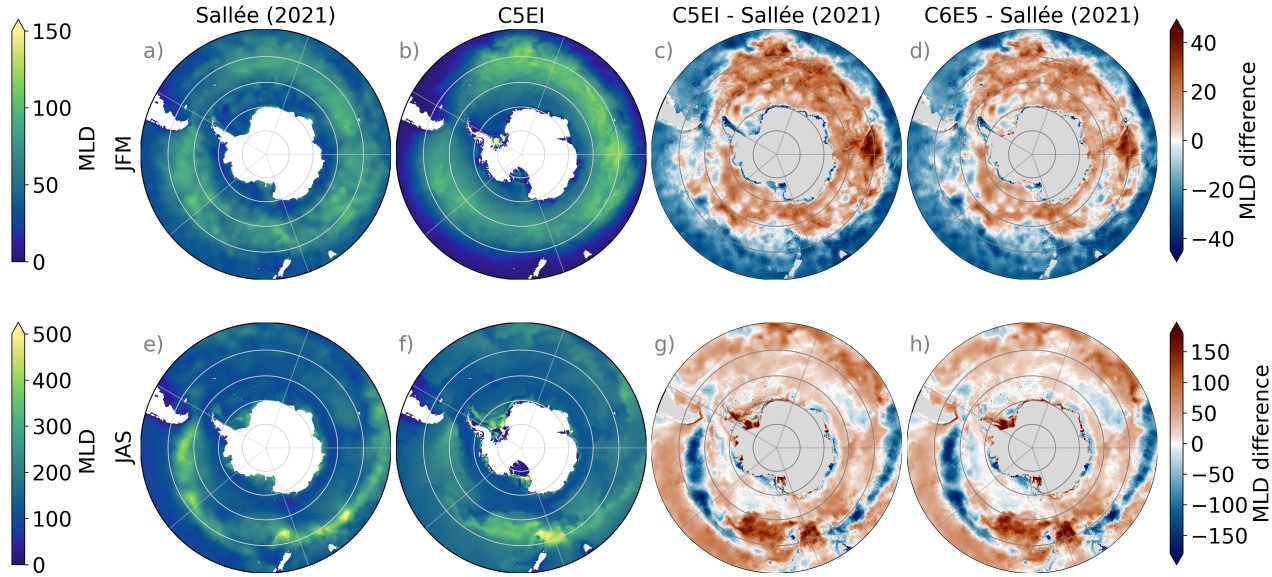

**Figure 12.** Summer (JFM) and winter (JAS) mixed layer depth (MLD) in meters. From left to right; Sallée et al. (2021a) climatology (a, e), C5EI simulations (b, f) and the difference between C5EI simulation and Sallée et al. (2021a) (c, g), and the C6E5 simulation and Sallée et al. (2021a) (d, h). The (Sallée et al., 2021a) climatology is calculated from observations 1970–2018 and the model runs are from 1992–2018. Note the different scales on colour bars for summer and winter.

which is strongest at the coast. The bug has been resolved in later versions of CICE, and we plan to address this issue in the next version of MetROMS-UHel.

### 4.2.2 Mixed layer depth

An accurate representation of the mixed layer depth (MLD) is crucial for reproducing air–sea exchanges and properly capturing the oceanic heat loss in winter. Regions of strong sea ice formation usually have wintertime deep water formation due to brine rejection. These processes have consequential effects on water mass properties. To validate the model runs, we compare monthly MLD values obtained from the simulations to the 50-year (1970–2018) observational MLD climatology (Sallée et al., 2021a), described in Sallée et al. (2021b). It is important to note that this dataset only overlaps with our experiments during the period 1992–2018. The MLD is calculated using a density ($\sigma$) criterion, as the depth at which the difference with the surface density is equal to 0.03 kg m$^{-3}$, following the same method of (Sallée et al., 2006, 2013).

The large scale patterns of the MLD are well reproduced (Fig. 12). In summer (JFM) the ACC ($\sim$ 60–50° S) is clearly recognizable as the area of deeper MLD (up to $\sim$ 100 m) (Fig. 12a, b). In this region, the modelled results tend to overestimate ($\sim$ 40 m) the MLD (Fig. 12c). On the other hand, the MLD is underestimated on the outer part of the domain (north of $\sim$ 50° S), as well as on most of the continental shelf. Changing the forcing from ERAI to ERA5 mitigates some of these biases. The MLD overestimations in the open ocean decrease with the updated atmospheric forcing, with smaller MLD in C5E5 than in

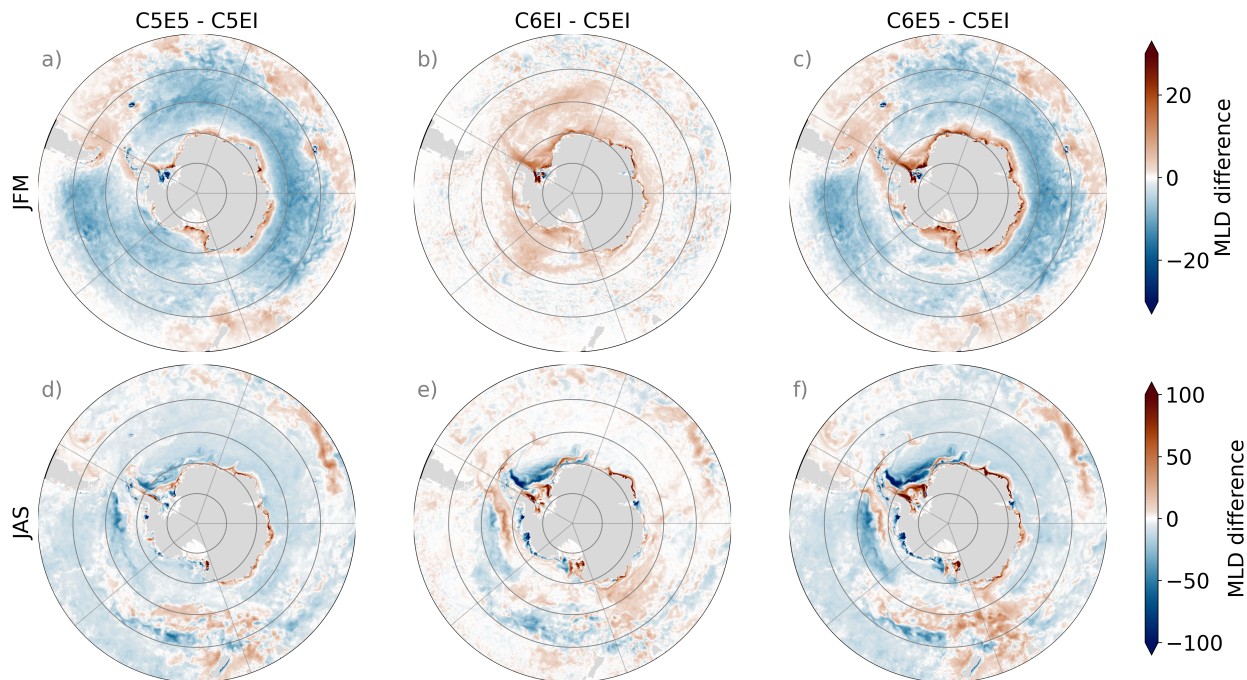

**Figure 13.** Summer (JFM) and winter (JAS) mixed layer depth (MLD) difference between models C5E5-C5EI (a, d), C6EI-C5EI (b, e) and C6E5-C5EI (c, f). Note the different scales on colour bars for summer and winter and different from Figure 12.

C5EI along the ACC (Figs 13a). This decreased MLD in the open ocean is likely due to the larger vertical salinity gradient at the surface, discussed in the previous section (Sect. 4.2.1) and shown in Fig. 11j. Furthermore, on the continental shelf, there is a decrease in the underestimation after the atmospheric forcing update, with deeper MLD values in C5E5 than in C5EI. Exceptions to this pattern occur under the Filchner-Ronne Ice Shelf in the Weddell Sea, where a strong shallowing of the MLD

is observed in C5E5 (Fig. 13a). The observed pattern of MLD increase on the continental shelf and decrease in the open ocean can probably also be linked to changes in wind fields between ERAI and ERA5 (Fig. B1e,f). Furthermore, the sea ice model update produces a small increase of the MLD (Fig. 12c, d and 13b), particularly in the southernmost areas of the domain. Therefore, the final setup, with both upgrades implemented, displays a small improvement of the summer (JFM) MLD with decreased MLD in the open ocean and increased MLD at the continental shelf (Fig. 13c).

Winter (JAS) MLD is larger than in summer, reaching depths over 400 m in some regions along the northern boundary of the ACC, through the BellAM, Ross Sea, and Pacific sectors, to the eastern part of the Indian Ocean (Fig. 12e). This pattern is also captured in C5EI (Fig. 12f), but the band of increased MLD is not as pronounced as in the observations. This is mostly caused by a general overestimation of MLD values over the domain, as well as by a seasonal signal over the ACC, where there is a negative bias and the MLD does not deepen as much as in the observation (Fig. 12g). The areas close to Tasmania and

New Zealand are an exception, where the ACC has a strong positive bias. The MLD underestimation could be explained by the lack of sinking AAIW in the models (Fig. 11i). The difficulty of preserving AAIW in MetROMS has earlier been discussed

in (Naughten et al., 2018b) and has been attributed to spurious diapycnal mixing with a potential contribution from errors in the surface forcing. The spurious diapycnal mixing and the drift in the density structure due to non-closure of the freshwater budget leads to a degradation of the Southern Ocean interior water masses, and to for example a weakening ACC (Fig. A1). The biases have a more complex pattern over the continental shelf, with mostly shallower MLD (Fig. 12g), and a few areas with high positive bias and possible excess deep water formation (e.g. next to the Ross, Filchner-Ronne, and Amery Ice Shelves).

Updating the atmospheric forcing from ERAI to ERA5 slightly decreases the wintertime MLD over most of the model domain, especially in the Weddell Sea sector at $\sim 70°$ S (just off the shelf) and in the BellAm sector a bit south of $60°$ S (Fig. 13d). In contrast, along the shelf, the update slightly increases the wintertime MLD, especially in the Weddell Sea, Indian Ocean, and Pacific Ocean sectors, mirroring the changes observed in summer. Updating the sea ice model from CICE5 to CICE6 results in a slightly deeper MLD along the shelves in most of the Weddell Sea, Indian Ocean, and western Ross Sea sectors, and a shallower MLD on the shelf in the east Ross Sea and BellAm sectors (Fig. 13e). There are no additional effects from updating both the atmospheric forcing and sea ice model (Fig. 13f).

### 4.3 Ice shelf basal melt

| Region | C5EI | C5E5 | C6EI | C6E5 | Adusumilli et al. | Rignot et al. |
|---|---|---|---|---|---|---|
| Filchner-Ronne | 64.8 | 66.7 | 61.8 | 63.5 | $81.4 \pm 123$ | $155.4 \pm 45$ |
| Eastern Weddell region | 85.3 | 70.9 | **94.1** | 83.8 | $156.1 \pm 63$ | $66.3 \pm 28$ |
| Amery | 92.7 (+) | 90.4 (+) | 104.6 (+) | 95.5 (+) | $45.6 \pm 40$ | $35.5 \pm 23$ |
| Australian sector | 33.9 (-) | 30.9 (-) | 36.8 (-) | 34.4 (-) | $171.2 \pm 54$ | $198.3 \pm 20$ |
| Ross Sea | **79.7** | **71.3** | **78.0** | **69.2** | $102.3 \pm 83$ | $70.1 \pm 34$ |
| Amundsen Sea | 83.7 (-) | 82.1 (-) | 105.6 (-) | 95.3 (-) | $310.8 \pm 43$ | $388.8 \pm 18$ |
| Bellingshausen Sea | 66.1 (-) | 61.5 (-) | 59.7 (-) | 50.4 (-) | $177.4 \pm 69$ | $187.2 \pm 31$ |
| Larsen Ice Shelves | **20.7** | **17.4** | **11.4** | **9.5** | $108.7 \pm 103.5$ | $22.1 \pm 68$ |
| Total Antarctica | 577.1 (-) | 537.4 (-) | 599.0 (-) | 545.1 (-) | $1264.3 \pm 147$ | $1325.0 \pm 235$ |

**Table 2.** The average ice-shelf basal mass loss (Gt $yr^{-1}$) for Antarctica is divided into 8 regions, following Naughten et al. (2018b). These regions encompass 25 ice shelves, as shown in Fig. B7. Model runs represent 1996–2018 averages. The melt rate is compared to Rignot et al. (2013) as acquired from Naughten et al. (2018b), and Adusumilli et al. (2020) data from their supplementary Table 1, meltwater flux 1994–2018 for the 25 ice shelves used in the analysis. The $(-)/(+)$ notation indicates values falling outside the range provided by Rignot et al. (2013) and Adusumilli et al. (2020), and **bold** values falls within both datasets. Notably, the flux in the table for Adusumilli et al. (2020) is not a steady state estimate and is slightly larger than the steady state estimate of $1100 \pm 60$ Gt $yr^{-1}$ for 1994–2018.

Basal melting of the Antarctic ice shelves is an important source of freshwater for the Southern Ocean. To validate the model results, we compare them to estimates of total meltwater flux from the basal melt of Antarctic ice shelves from 1994 to 2018, as provided by Adusumilli et al. (2020) and to data from Rignot et al. (2013) acquired from Naughten et al. (2018b). Assuming

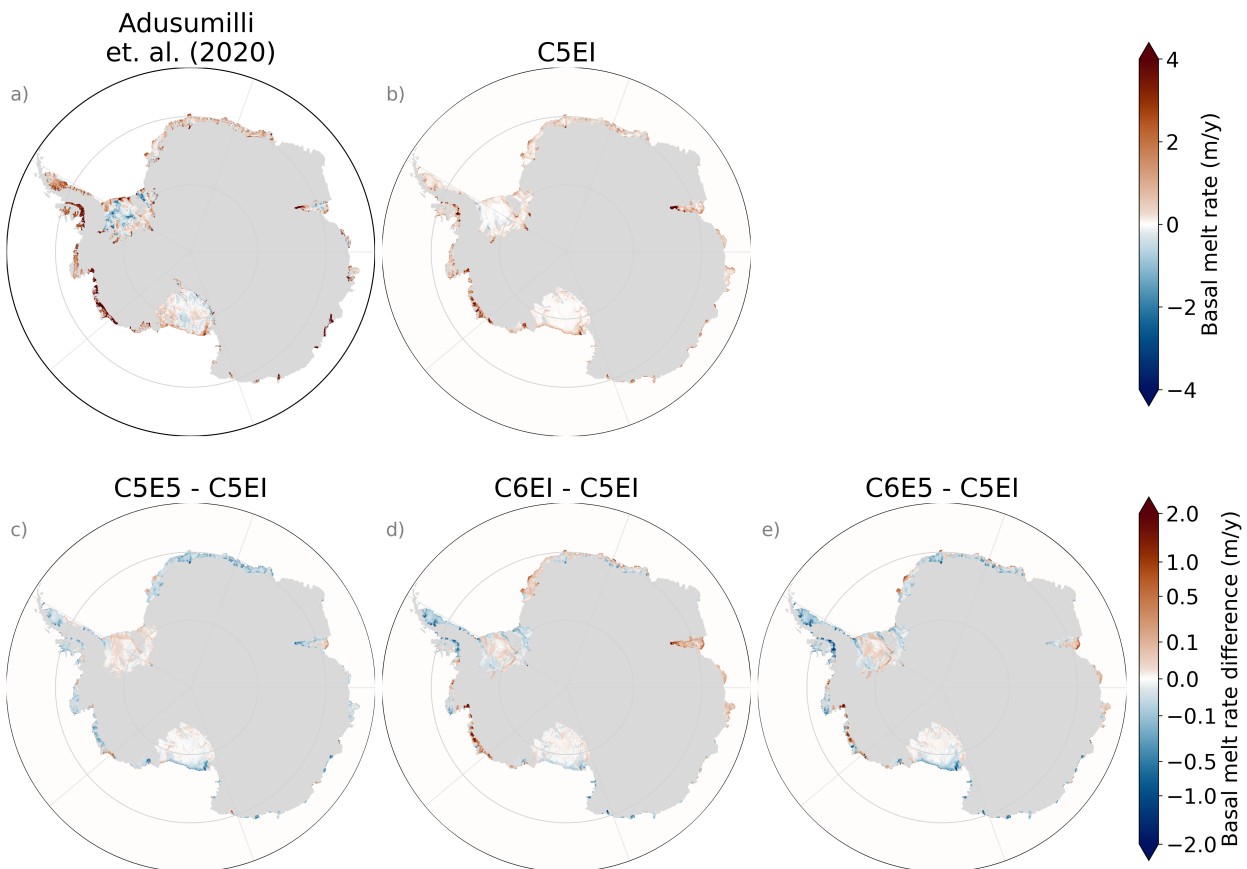

**Figure 14.** Map of annual average melt rate (m yr$^{-1}$) for (a) observational estimate from Susheel Adusumilli et al. (2020), (b) C5EI model run, and (c–e) difference between the C5EI model run and the other model runs. The model run averages are calculated from 1996–2018. Note that differences have a different colour scale.

a steady state, Adusumilli et al. (2020) calculated a total flux of 1100±60 Gt yr$^{-1}$ for 1994–2018. These estimates were done using CryoSat-2 radar altimetry, satellite-derived ice velocities, a model of surface mass balance, and firn state variability.

Our model runs basal melt rates reach equilibrium in 1996 (Fig. B8), and thus the average melt rates for the model simulations are calculated over the 1996–2018 period. All the model runs underestimate the total ice shelf basal mass loss compared to observations by, approximately, a factor of two. The simulated loss averaged over 1996–2016 is 577 Gt yr$^{-1}$ for C5EI, 537 Gt yr$^{-1}$ for C5E5, 599 Gt yr$^{-1}$ for C6EI and 545 Gt yr$^{-1}$ for C6E5 (Table 2). Updating the atmospheric forcing from ERAI to ERA5 results in a decrease in the melt, while updating the sea ice model leads to a slight increase.

The spatial distribution of the melt bias is regionally dependent (Fig. 14). For instance, in the Filchner-Ronne Ice Shelf melting occurs at both the front and the grounding line, with refreezing in the center and western regions. The model underestimates refreezing in the middle and significantly underestimates melting at the front. However, the average mass loss simulated for

the Filchner-Ronne Ice Shelf is within the range of that estimated by Adusumilli et al. (2020), but below the range reported by Rignot et al. (2013) (Table 2). A similar, but smaller, underestimation of refreezing and melting can be seen in the Ross Ice Shelf (Fig. 14). Modeled values of the Ross Ice Shelf area is within the range of both datasets. However, discrepancies between Adusumilli et al. (2020) and the model runs arise for the smaller ice shelves in the region, such as Nickerson and Sulzberger Ice Shelves (not shown). The largest underestimations, compared to both observational datasets, occur in the Australian region as well as the Amundsen and Bellingshausen Sea regions (Table 2). In these regions, all the models show values far below the range given by Rignot et al. (2013), except in the Wilkins Ice Shelf in the Bellingshausen Sea, where uncertainty is large. When comparing to Adusumilli et al. (2020), the Wilkins and Abbot Ice Shelves in the Bellingshausen Sea and many of the ice shelves in the Australian region is within the given range (not shown) due to a large uncertainty. However both regions as a whole are clearly outside the uncertainty ranges of Adusumilli et al. (2020) for these regions (Table 2). In the BellAm sector, this is likely due to the warm shelf area being too cold in the model runs (Fig. B6b), while this is not the case for the Australian region, which is warmer than observations in the model runs (Fig. B6b). Naughten et al. (2018b) also underestimated melt in the Australian region and speculate that it is due to a lack of HSSW. The Amery Ice Shelf is the only region where the model runs overestimate basalt melt compared to the observations (Table 2), with most of the loss concentrated at the grounding line. A similar overestimation was reported in the MetROMS-Iceshelf simulations of Naughten et al. (2018b).

Updating the atmospheric forcing from ERAI to ERA5 (Fig. 14c) generally reduces melt rates, except in the Filchner-Ronne region (Table 2). This reduction of melt rates is related to the generally colder ocean in C5E5 than in C5EI south of 60° S (Fig 11d). This colder ocean is, in turn, likely a consequence of stronger ERA5 coastal winds compared to ERAI (Fig. B1e and f), as discussed in Sect. 4.2.2, resulting in a stronger oceanic mixing (Fig. 13a and d), and heat loss to the atmosphere cooling the upper ocean.

The transition from CICE5 to CICE6 (Fig. 14d) results in varying regional melt rate responses. Melt rate increases are observed especially under the Amery Ice Shelf and over the Amundsen Sea (Table 2), with the largest increase over the Amundsen Sea. In these regions the temperature in the ice shelf cavities has increased (not shown), associated with shallower coastal C6EI winter mixed layers (Fig. 13e) and higher sea ice concentration (Fig. B2k) than in C5EI, indicating reduced oceanic heat loss to the atmosphere. In contrast, melt rates decrease on both sides of the Antarctic Peninsula, in the Bellingshausen Sea and the Riisen-Larsen Sea, as well as under the Filchner Ice Shelf and the Ross Sea region (Table 2). However, these decreases are not as regionally consistent as the melt rate increases (Fig. 14d), but could be related to deeper winter mixed layers (Fig. 13e), increased oceanic heat loss and colder water masses on the continental shelf. When considering the combined effects of ERA forcing and CICE model updates, the melt rate changes reflect the influence of both individual updates (Fig. 14e, Table 2).

The fact that all MetROMS-Iceshelf simulated melt rates are, on average, lower than the observed, have been discussed in Naughten et al. (2018b), sections 4.3 and 4.3.6. In addition to the previously discussed hydrographic biases, the absence of tidal-driven mixing and insufficient eddy heat transport - due to inadequate model resolution - can contribute to reduced basal melt. Notably, the relative importance of these factors varies regionally, for example, tides might play a significant role in the Filchner-Ronne Ice Shelf. Finally, little is known of ice-shelf basal roughness, the associated drag and turbulent transfer

coefficients, and their temporal and spatial variability. As Naughten et al. (2018b) emphasize, these ice-shelf/ocean interaction parameters should be better constrained through observations.

## 5 Conclusions

We present the MetROMS-UHel ocean–sea ice model, and assess how the modelling results have changed with the updated ERA atmospheric forcing and the sea ice model CICE compared to the MetROMS-Iceshelf model. Overall, the new model has a better representation of the sea ice than the previous version. Specifically, the MetROMS-Iceshelf model underestimates the sea ice concentration and area compared to observations, but in the MetROMS-UHel model this negative bias is reduced. For the average sea ice volume, where no observations are available for comparison, the model shows increases with the ERA 585 update, but decreases with the CICE update for a very small combined effect. Both model versions capture the sea ice drift patterns well, but the vector correlation shows a slight improvement in MetROMS-UHel by both the ERA and the CICE update.

In the ocean, both model versions are able to identify the main water masses on the shelf, though with a positive bias in interior ocean temperatures (compared to observations). Furthermore, MetROMS-iceshelf overestimates the MLD in the deep ocean while underestimating MLD on the shelf. The ERA update slightly reduces these MLD biases. The CICE update, on the 590 other hand, causes a clear decrease in coastal salinity, resulting in a negative salinity bias in MetROMS-UHel. Adjusting the configuration of the coupled ROMS–CICE6 model under the ERA5 forcing is required to mitigate this effect (Barthélemy et al., 2018). In the ice shelf cavities, MetROMS-iceshelf underestimates melt rates by approximately a factor of two. In MetROMS-UHel the ERA update increases this underestimation, while the CICE update slightly decreases it. However, the effect of these model updates is comparably small and regionally varying.

Overall, both the atmospheric forcing and the sea ice model update improve the model system. The ERA update has a stronger impact on sea ice area, concentration, and basal melt rate, while the CICE update has a stronger effect on the sea ice volume, drift, and the ocean hydrography. However, the amount and direction of changes vary regionally, suggesting that the importance of physical processes and external forcing depends on the region.

## Appendix A: Drake Passage Transport analysis

The Drake Passage Transport (DPT) is used to estimate the ACC strength. Recent observational DPT estimates are based on current meter arrays. Specifically, the DRAKE mooring array (Koenig et al., 2014), that gives a full-depth transport of $141\pm2.7$ Sv, and the cDrake mooring Array (Chidichimo et al., 2014; Donohue et al., 2016) with a DPT of $173.3\pm10.7$ Sv. Both of these are larger than the older canonical estimate of 134 Sv (Whitworth and Peterson, 1985; Cunningham et al., 2003).

    The DPT timeseries from the model runs have been estimated at 67° W and can be seen in Fig. A1. All the runs underestimate
the DPT compared to the observations and have a clear decreasing trend during the analysis period. Both model configuration updates (the CICE6 sea ice model and ERA5 atmospheric forcing) cause a slight decrease in the transport, with C5EI showing the highest 1992–2018 mean value at 122.5 Sv and C6E5 the smallest with 111.3 Sv. The CICE6 runs show a decreasing trend of -1 Sv/yr, over the analysis period. However, the calculated DPT for the later part of the simulation (2005–2018) suggests that the decreasing trend slows down during the second part of the CICE6 analysis period, but not in the CICE5 runs.

Nevertheless, this study is focused on the surface ocean and on relative short time scales, so the negative trend in DPT which occurs in the deeper ocean should not substantially affect the results presented in this work.

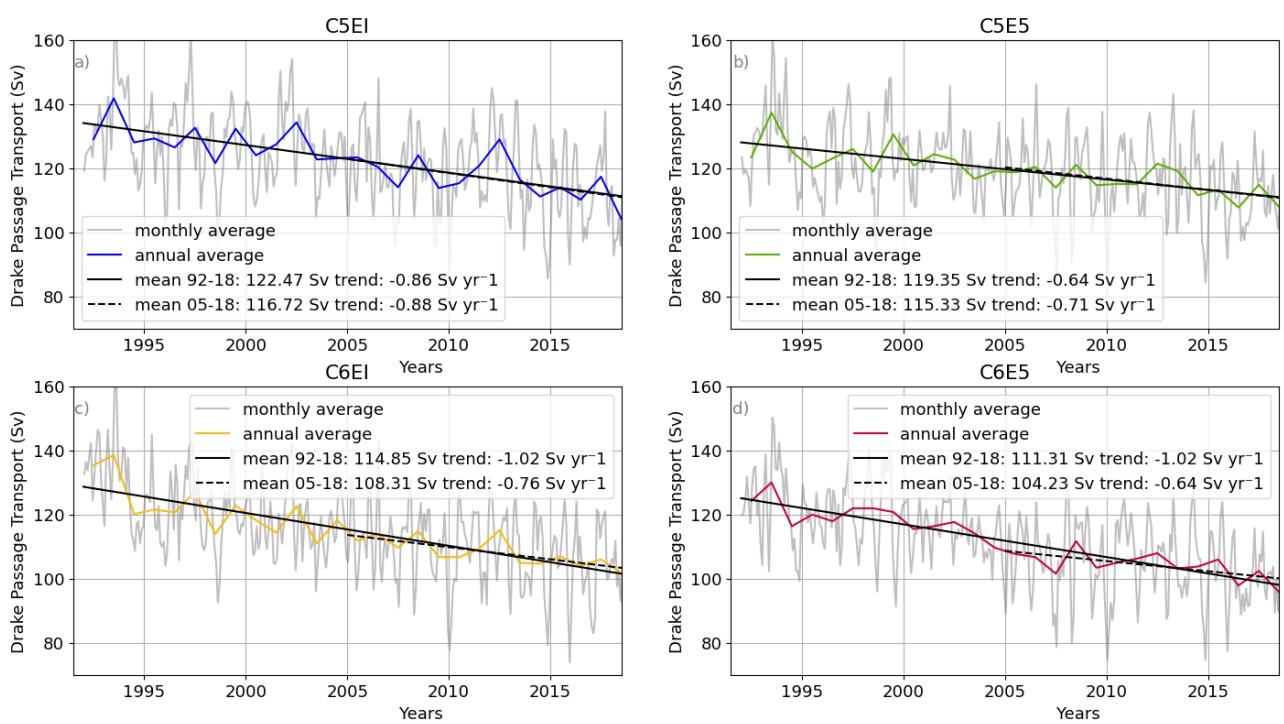

**Figure A1.** Drake Passage Transport (DPT) for all model runs. Gray lines show monthly averages, while the coloured lines show the annual averages for each model. Linear fits for the whole period (1992–2018) are shown in solid black lines; linear fits for the last half of the analysed period (2005–2018) are shown in black dashed lines

# Appendix B: Additional figures and tables

| Variable | 1992-2014 Mean | 1992-2014 Slope | 2014-2018 Mean | 2014-2018 Slope | 1992-2018 correlation to NSIDC |
|---|---|---|---|---|---|
| **February** | | | | | |
| C5EI | 0.90 | 0.000 | 0.88 | -0.121 | 0.490 |
| C5E5 | 1.12 | 0.004 | 1.17 | -0.148 | 0.683 |
| C6EI | 0.86 | -0.001 | 0.78 | -0.136 | 0.471 |
| C6E5 | 1.18 | 0.001 | 1.12 | -0.193 | 0.655 |
| Ice Index | 2.05 | 0.025 | 2.06 | -0.294 | 0.980 |
| NSIDC | 2.23 | 0.021 | 2.17 | -0.316 | - |
| **September** | | | | | |
| C5EI | 15.02 | 0.017 | 14.44 | -0.435 | 0.801 |
| C5E5 | 15.38 | 0.004 | 14.73 | -0.442 | 0.804 |
| C6EI | 15.38 | 0.033 | 14.80 | -0.532 | 0.840 |
| C6E5 | 15.70 | 0.018 | 15.07 | -0.566 | 0.868 |
| Ice Index | 14.74 | 0.036 | 14.44 | -0.363 | 0.960 |
| NSIDC | 16.56 | 0.035 | 16.18 | -0.384 | - |

**Table B1.** Mean and slope values for timeseries for sea ice area [$10^6$ km$^2$] in February and September (Fig.4): 1992-2014 and 2014-2018 as well as the Pearson correlation calculated against NSIDC for 1992-2018

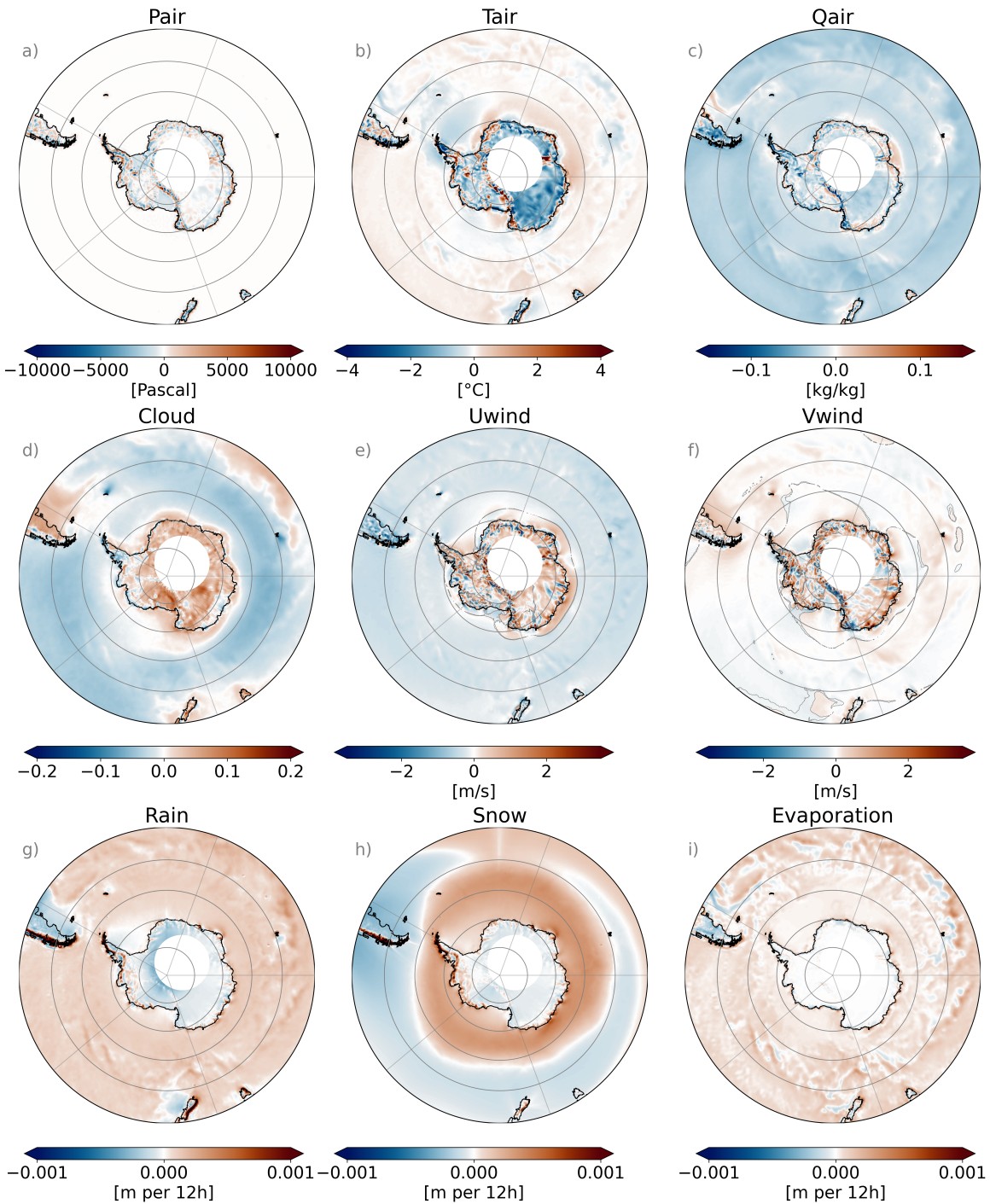

**Figure B1.** Differences between ERA5 and ERAI forcing fields calculated from 1992-2018 average. For Uwind and Vwind the plots (e,f) shows the change of the strength of the wind component in its direction calculated as $wind\_diff = (wind\_era5/abs(wind\_era5)) * (wind\_era5 - wind\_eraI)$, and the gray lines indicates the areas where the direction of the wind vector flips into the opposite direction.

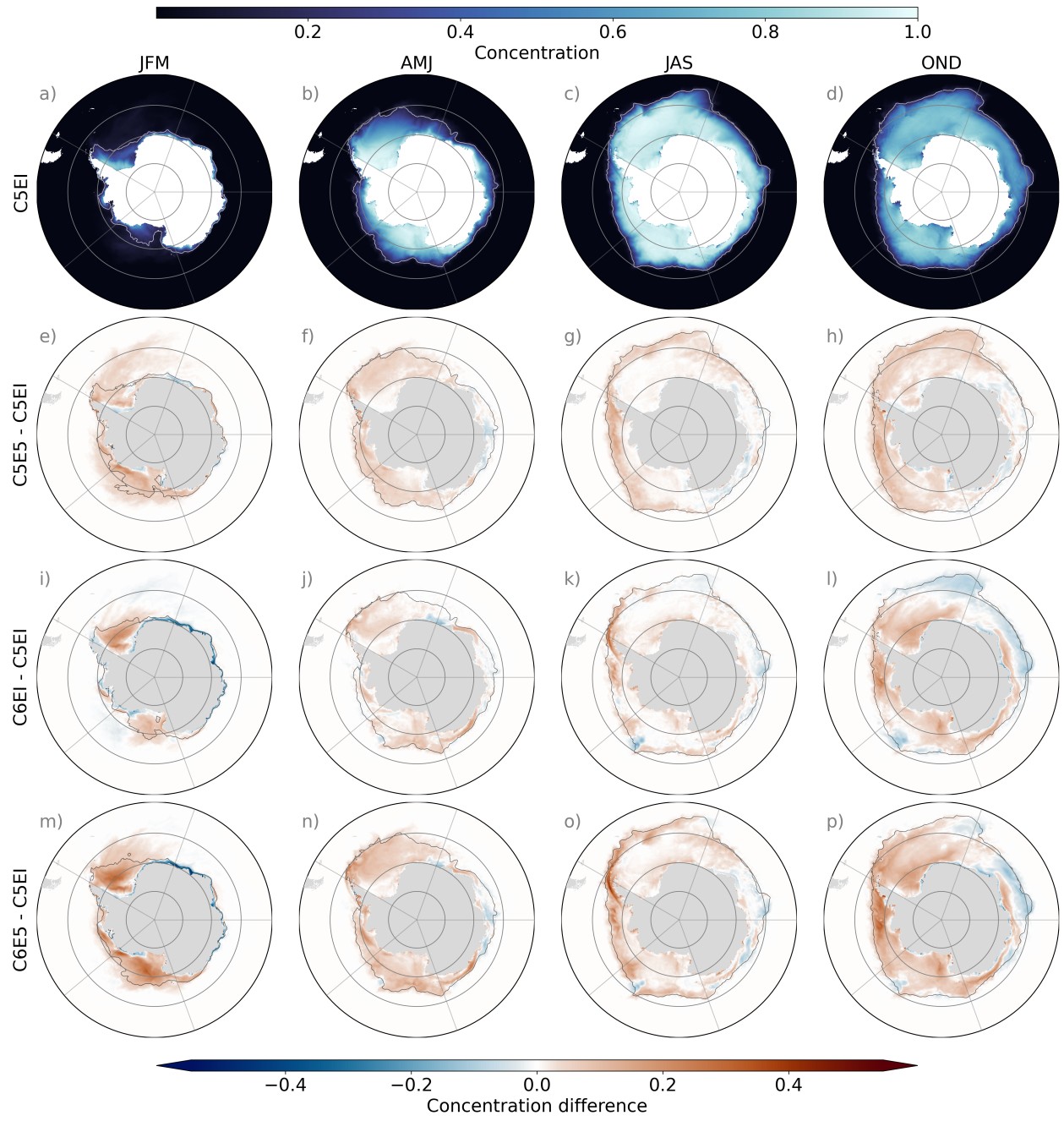

**Figure B2.** Sea ice seasonal mean concentration over the period 1992–2018 from model run C5EI (a–d) and concentration differences (e–p) between C5EI and the other model runs (C5E5, C6EI, C6E5). The black (white) line shows the sea ice extent.

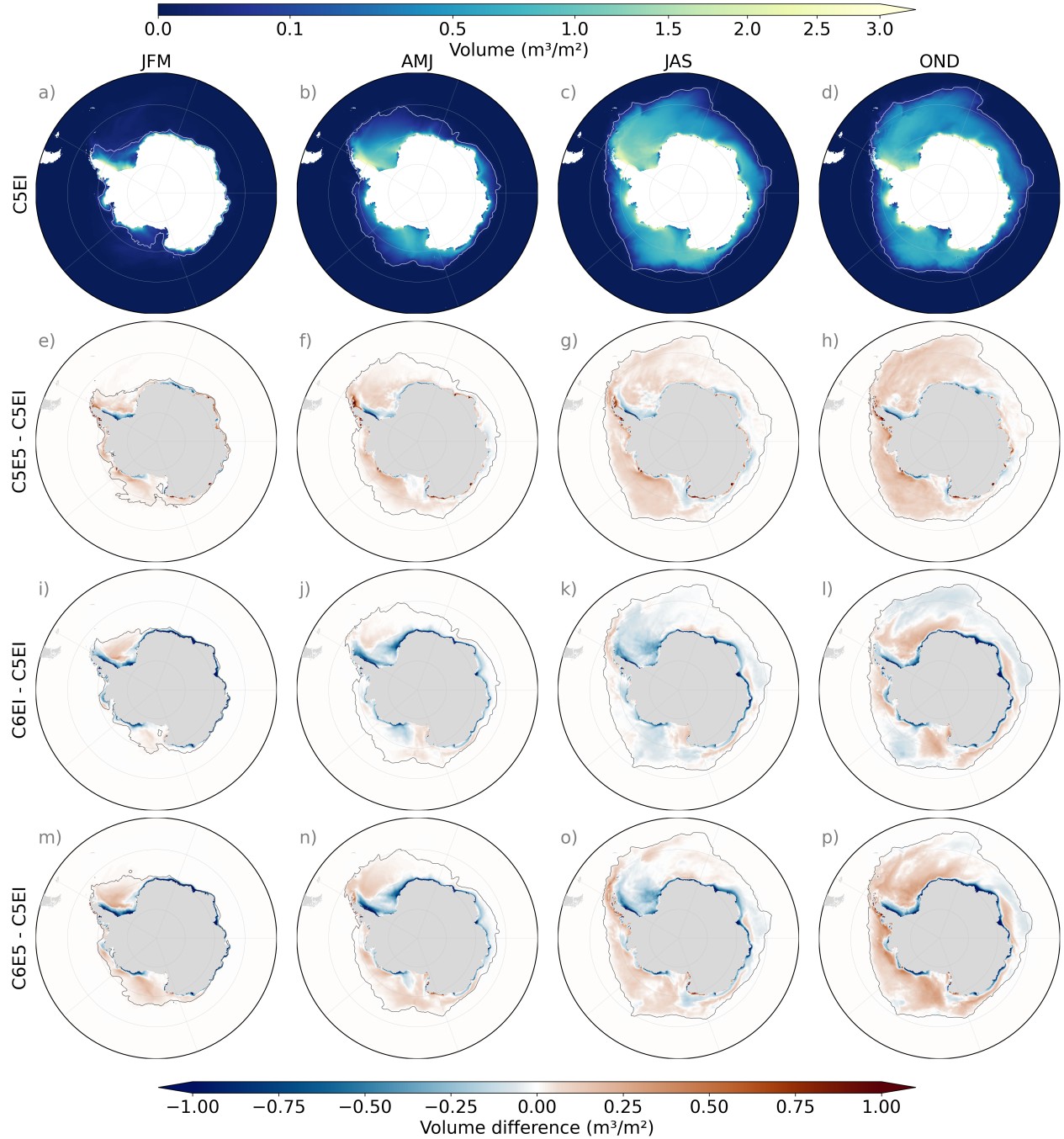

**Figure B3.** Sea ice seasonal mean volume [m$^3$/m$^2$] over 1992–2018 from model run C5EI (a–d) and volume differences (e–p) between C5EI and the other model runs (C5E5, C6EI, C6E5). The black (white) line shows the sea ice extent.

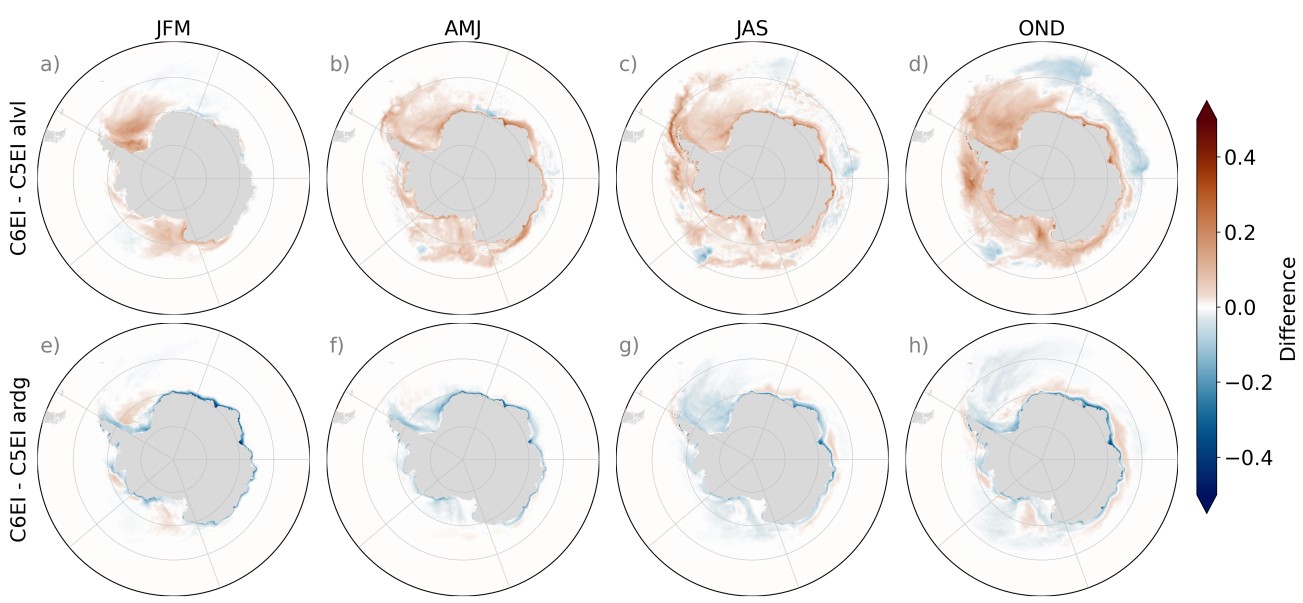

**Figure B4.** Difference of level ice area fraction (alvl) (a–d) and ridged ice area fraction (ardg) (e–h) between the C6EI and C5EI simulations. Values are seasonal means from 1992–2018.

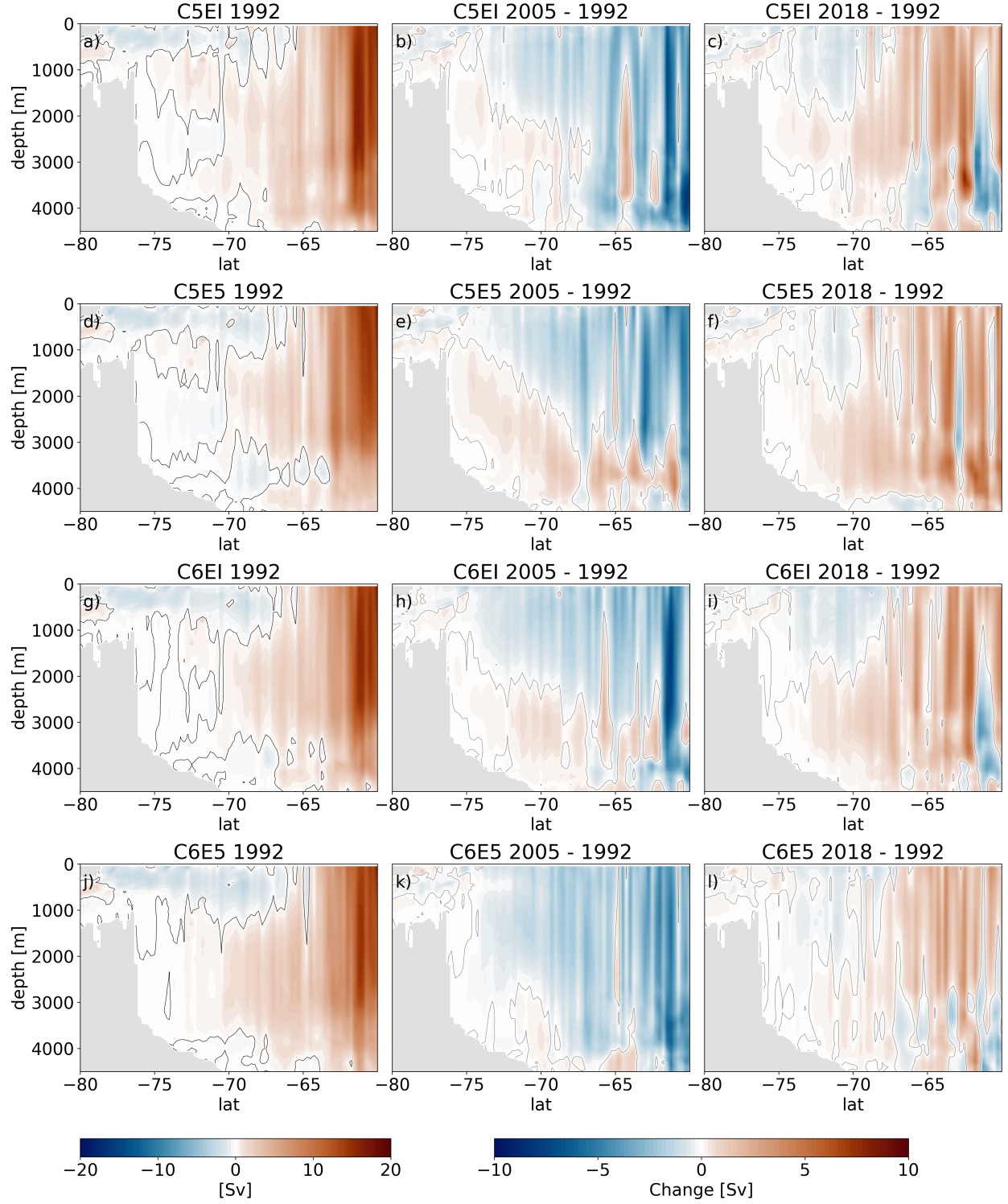

**Figure B5.** Overturning streamfunction for January 1992 (a,d,g,j), and change from January 1992 to January 2005 (b,e,h,k) and 2018 (c,f,i,l) for all model runs south of $60°$S

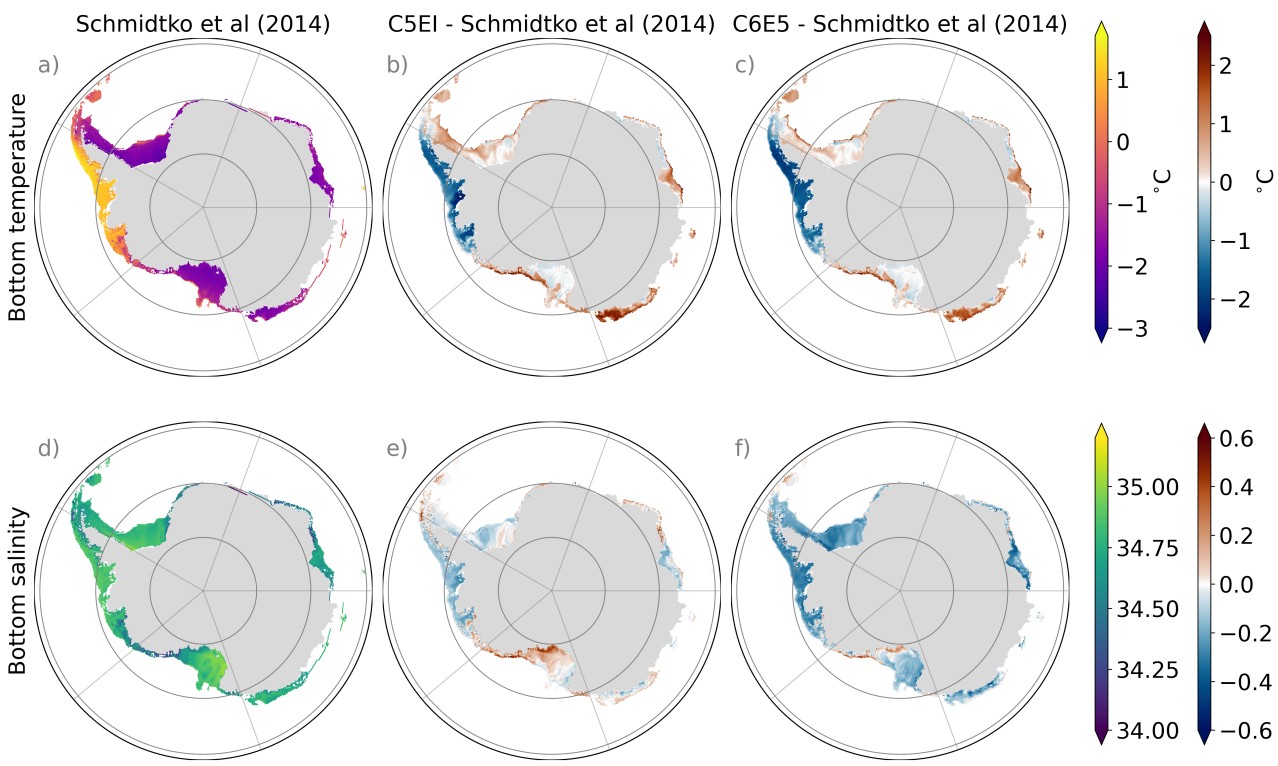

**Figure B6.** (a) Annual average bottom potential temperature (°C), (d) salinity, both from Schmidtko et al. (2014), and (b, e) the difference between the model run C5EI and Schmidtko et al. (2014), and (c, f) C6E5 and Schmidtko et al. (2014). The period considered is 1992–2012. The absolute salinity for the model runs was calculated using the TEOS-10 standard.

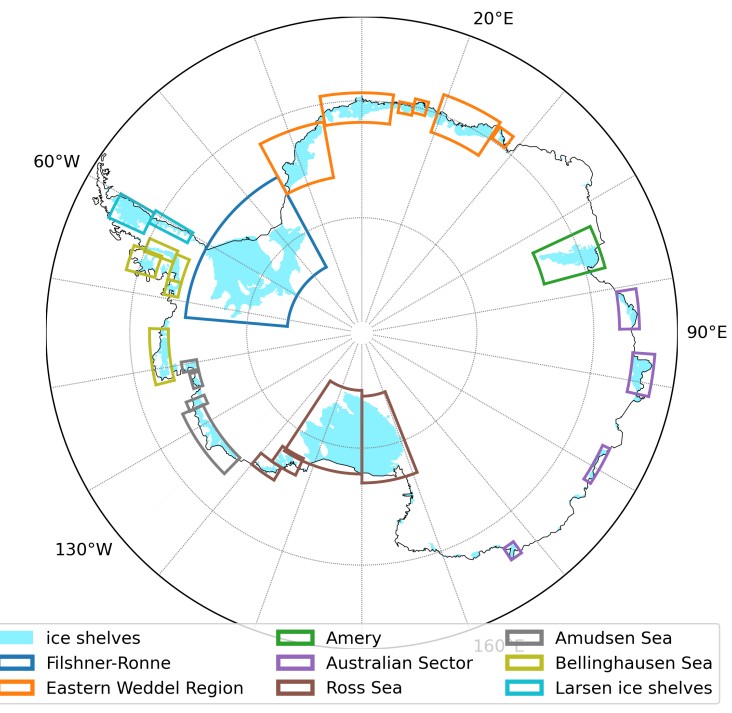

**Figure B7.** Figure of the bounding boxes used for the ice shelves for the comparison in Table 2

.

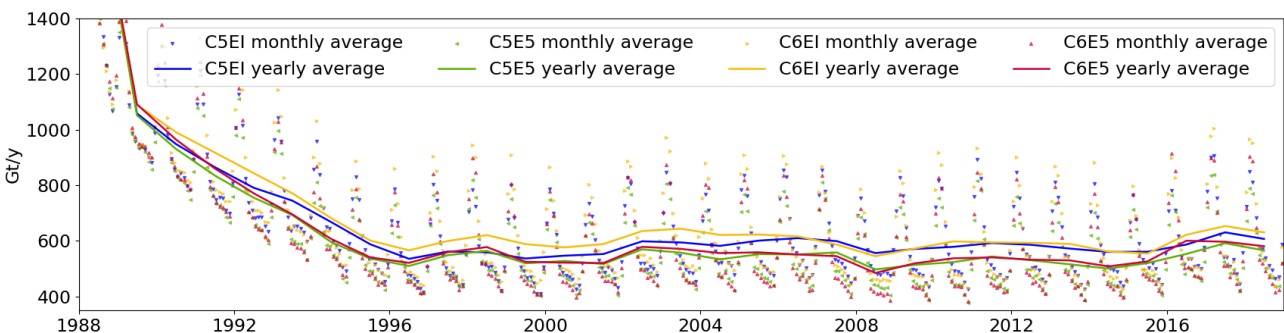

**Figure B8.** Timeseries of melt rate from the different model runs C5EI (blue), C5E5 (yellow), C6EI (green), and C6E5 (red) from the start of the spinup. The years 1988 to 1991 stand for the spinup with the year 1992 forcing. Solid lines show yearly averages and dots show monthly averages. The models take until 1996 to find a balance.

*Code and data availability.* MetROMS-Iceshelf model code is available at https://doi.org/10.5281/zenodo.1157229 (Naughten et al., 2018a), MetROMS-UHel model code is available at https://doi.org/10.5281/zenodo.14185734 (Äijälä and Uotila, 2024) and code for analyzing the model output as well as relevant model data to produce the figures can be found at https://doi.org/10.5281/zenodo.15356488 (Äijälä and Nie, 2024).

*Author contributions.* **Cecilia Äijälä:** Conceptualization, Methodology, Software, Validation, Formal analysis, Investigation, Data Curation, Writing - Original Draft, Writing - Review & Editing, Visualization. **Yafei Nie:** Methodology, Software, Validation, Formal analysis, Writing - Original Draft (Sea Ice Drift), Writing - Review & Editing, Visualization. **Lucía Gutiérrez-Loza:** Conceptualization, Writing - Original Draft, Writing - Review & Editing. **Chiara De Falco:** Conceptualization, Writing - Original Draft, Writing - Review & Editing. **Siv Kari Lauvset:** Conceptualization, Writing - Original Draft, Writing - Review & Editing, Funding acquisition **Bin Cheng:** Writing - Review & Editing **David A. Bailey:** Writing - Review & Editing **Petteri Uotila:** Conceptualization, Resources, Writing - Original Draft, Writing - Review & Editing, Supervision, Project administration, Funding acquisition.

*Competing interests.* The authors declare that they have no conflict of interest.

*Acknowledgements.* This study was undertaken as part of the EU Horizon 2020 PolarRES project (https://polarres.eu), funded under grant agreement number: 101003590. The authors thank the EU Commission for facilitating this research. Additionally, Yafei Nie was supported by the China Postdoctoral Science Foundation (Grant No. 2023M741526) The authors wish to acknowledge CSC – IT Center for Science, Finland, for computational resources. EN.4.2.2 data were obtained from https://www.metoffice.gov.uk/hadobs/en4/ and are © British Crown Copyright, Met Office, 2024, provided under a Non-Commercial Government Licence http://www.nationalarchives.gov.uk/doc/non-commercial-government-licence/version/2/. The Metroms-iceshelf code was acquired from https://github.com/knaughten/metroms_iceshelf and input files for the circumpolar setup were acquired from The University of Tasmania. Some of the code for analyzing and plotting the data is based on Kaitlin A. Naughten's roms_tools https://github.com/knaughten/roms_tools, and Fabio Boeira Dias' waom notebook https://github.com/fabiobdias/waom_notebook. ChatGPT was used to write some of the code for plotting the figures and Grammarly was used to check the language of parts of the manuscript.

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
