# Peer review of "Impacts of CICE sea ice model and ERA atmosphere on an Antarctic MetROMS ocean model, MetROMS-UHel-v1.0"

_Geoscientific Model Development, 2024_

## Author Comment (AC1)

**Response to Referee Comments**

March 7, 2025

The document is color coded in the following way:
Referee comments are in **blue**.
Our answers are in **black**.
Citations from the revised manuscript are in **red**.

**Anonymous Referee #1**

"Impacts of CICE sea ice model and ERA atmosphere on an Antarctic MetROMS ocean model, MetROMS-UHel-v1.0" by Äijälä and colleagues evaluates a circumpolar Southern Ocean model including sea ice and ice shelf cavities. Compared to a previously published version of the model, both the atmospheric forcing and sea ice component have been updated. The impacts of each change in isolation are evaluated, as well as both changes together. Evaluation focuses on sea ice properties, with ocean hydrography and ice shelf melting given a secondary focus. In general the updates to the model are beneficial for the representation of sea ice, but problematic for ocean hydrography, particularly a freshening of the continental shelves.

**General comments:**

In general this is a solid and thorough evaluation paper including some clever analysis of sea ice model performance. I believe it is suitable for GMD after moderate revisions.

We would like to thank the reviewer for their helpful and constructive comments. We address your inquiries below in the best way possible.

In particular, there are a few missing pieces which would help to fill in the oceanographic gaps in the story:

1. Some quantification of ACC strength, ideally a figure showing transport through Drake Passage. The text mentions it slows down, which is concerning, but does not quantify either the initial bias or the degree of drift.

Thank you for the suggestion, we agree with Reviewer #1 that the decreasing trend of the Drake Passage Transport (DPT) is concerning. An assessment of the DPT has been added to the manuscript in Appendix A, including Fig. 1 (Fig. A1 in the revised manuscript) showing the DPT for all model runs including the linear trends. In the Appendix, on lines 605–617, a brief description of the DPT measurements and the model output results regarding ACC strength have been included.

2. A figure of the overturning streamfunction in each model configuration. I suspect that the coastal freshening induced by the update to CICE6 is shutting down dense water export from the shelf. Quantifying the strength of the lower cell would confirm this.

We have added a figure of the overturning streamfunction from 1992, 2005 and 2018 (Fig. 2) to the Appendix B as Figure B5 and added a paragraph discussing this figure to Sect. 4.2.1 lines 439–446.

A possible consequence of this salinity bias is that the freshening of coastal waters induced by the update to CICE6 disrupts the export of dense water from the shelf, potentially shutting it down. Figure B5 presents the streamfunction values north of 60° S, between 3000 and 5000 m depth. The lower

[Figure]

Figure 1: Fig. A1 in the manuscript: Drake Passage Transport (DPT) for all model runs. Gray lines show monthly averages, while the coloured lines show the annual averages for each model. Linear fits for the whole period (1992–2018) are shown in solid black lines; linear fits for the last half of the analysed period (2005–2018) are shown in black dashed lines

overturning cell, identified by negative values south of 50° S near Antarctica, initially exhibits a higher intensity in CICE6 runs compared to CICE5 in January 1992 (Fig. B5a, d, g, j). From 1992 onward, this cell weakens across all simulations, indicating a slowdown in dense water export. Between 1992 and 2005, the weakening is significantly more pronounced in CICE6 than in CICE5, as evidenced by the stronger anomaly in the C6EI/C6E5 2005–1992 difference (Fig. B5h, k) compared to C5EI/C5E5 (Fig. B5b, e). This trend persists in the 2018–1992 comparison (Fig. B5c, f, i, l), suggesting a lasting impact of the CICE6 update on deep water export.

3. Some analysis of the sea ice formation rates in each experiment, as this process is crucial for setting the shelf water masses.

Thank you for the suggestions, we plotted both seasonal and annual growth and melt. The annual average growth and melt, as well as the difference of these, can be seen in Fig. 3, and it has been added as Fig. 7 to the manuscript. The figure results are discussed in a new subsection (Sect. 4.1.4), lines 341–379.

4. A figure (bar chart?) showing ice shelf mass loss integrated over each main ice shelf or region, compared to observations. This would make it much easier to gauge the regional dependence in model bias. From the existing figures it is very difficult to parse the magnitude of the bias or model sensitivity in the small, high-melt cavities in the Amundsen Sea, as well as the large, cold cavities which experience both melting and refreezing.

We have added Table 1 to Sect. 4.3 and Fig. 4 to Appendix B of the revised manuscript as Table 2 and Fig. B7, respectively. We use the same regional division as Naughten et al. (2018b), which defines and analyses eight regions comprising 25 ice shelves. For observations, we present values derived from both Susheel Adusumilli et al. (2020) and Rignot et al. (2013). We have reworked much of the Sect. 4.3, lines 532–580, to incorporate this table and answer specific questions of both reviewers.

The manuscript also errs too much on the side of simple description, with some missed opportunities

[Figure]

Figure 2: Fig. B5 in the manuscript: Overturning streamfunction for January 1992 (a,d,g,j), and change from January 1992 to January 2005 (b,e,h,k) and 2018 (c,f,i,l) for all model runs.

[Figure]

Figure 3: Fig. 7 in the manuscript: Mean sea ice growth (frazil ice growth+congelation ice growth+snow ice formation) and melt (top ice melt + basal ice melt + lateral ice melt) for C5EI (a, e) and difference between the other models and C5EI (b–c, f–h) during 1992–2018. The last row (i–l) shows the difference between the growth and melt above, for C5EI (i) and the model differences (j–k).

for attribution (i.e. which physical processes are behind the model bias visualised?) Of course it would be excessive to fully analyse the causes of all model biases in one manuscript, but at least some discussion of the likely possibilities is warranted. In my specific comments below I have highlighted examples where this is particularly needed.

Thank you, we agree, please see our response to the specific comments below.

**Specific Comments:**

Line 5: It's worth mentioning more explicitly in the abstract that ice shelf cavities are included, as this is not generally true of Southern Ocean models.

We have added this mention to the abstract, in Line 6–7:

The two versions of MetROMS evaluated in this study use a version of the regional ROMS ocean model including ice shelf cavities.

| Region | C5EI | C5E5 | C6EI | C6E5 | Adusumilli et al. | Rignot et al. |
|---|---|---|---|---|---|---|
| Filchner-Ronne | 64.8 (-) | 66.7 (-) | 61.8 (-) | 63.5 (-) | 65.5 (-) | $155.4 \pm 45$ |
| Eastern Weddell region | 85.3 | 70.9 | 94.1 | 83.8 | 154.3 (+) | $66.3 \pm 55$ |
| Amery | 92.7 (+) | 90.4 (+) | 104.6 (+) | 95.5 (+) | 50.5 | $35.5 \pm 23$ |
| Australian sector | 33.9 (-) | 30.9 (-) | 36.8 (-) | 34.4 (-) | 114.8 (-) | $198.3 \pm 36$ |
| Ross Sea | 79.7 | 71.3 | 78.0 | 69.2 | 98.8 | $70.1 \pm 39$ |
| Amundsen Sea | 83.7 (-) | 82.1 (-) | 105.6 (-) | 95.3 (-) | 307.6 (-) | $388.8 \pm 33$ |
| Bellingshausen Sea | 66.1 (-) | 61.5 (-) | 59.7 (-) | 50.4 (-) | 169.1 | $187.2 \pm 59$ |
| Larsen Ice Shelves | 20.7 | 17.4 | 11.4 | 9.5 | 100.5 | $22.1 \pm 81$ |
| Total Antarctica | 577.1 (-) | 537.4 (-) | 599.0 (-) | 545.1 (-) | 1209.9 | $1325.0 \pm 235$ |

Table 1: Table 2 in the manuscript: The average ice-shelf basal mass loss ($Gt\ yr^{-1}$) for Antarctica is divided into 8 regions, following Naughten et al. (2018b). These regions encompass 25 ice shelves, as shown in Fig. B7. Model runs represent 1996–2018 averages, while Susheel Adusumilli et al. (2020) dataset provides an average melt rate for 2010–2018. The melt rate is compared to Rignot et al. (2013) as acquired from Naughten et al. (2018b). The $(-)/(+)$ notation indicates values falling outside the range provided by Rignot et al. (2013). Notably, the 2010–2018 estimate of the 1209.9 $Gt\ yr^{-1}$ estimated by Susheel Adusumilli et al. (2020) differs slightly from the steady state value of $1100 \pm 60\ Gt\ yr^{-1}$ for 1994–2018 reported by Adusumilli et al. (2020).

Line 37: You could make the argument that modelling Antarctic sea ice is also challenging because historically, most development and tuning has been focused on Arctic conditions, which are not entirely transferable to Antarctica.

Thank you, this is a good argument. We have updated the text in Lines 47–48:

Model development and tuning have historically been focused on Arctic sea ice, and such efforts are not entirely transferable to Antarctica, making the modeling of Antarctic sea ice more challenging.

Line 68: What specific changes have been made to the CICE physics parameterisations? If we knew which processes in particular had been updated or added, it could make attribution of the model behaviour easier later. Were any of the bug fixes critical?

The transition from CICE5.1.2 to CICE6.0 was a major refactoring of the code and a few bug fixes. However, with the introduction of the quality control test (QC) as documented in Roberts et al. (2018) these bug fixes and refactoring were not considered 'climate changing', but for example a change in the salinity and fresh water flux calculations states in the release notes (Elizabeth Hunke, 2018) that it might change results in ocean coupled simulations. Additional changes from CICE6.0 to CICE6.3 were also not climate changing and most of these impacted the non-default physics, such as the floe size distribution.
This is now mentioned shortly in the text at Lines 74–79, that now reads:

CICE has been completely reworked between versions 5 and 6, with major restructuring and refactoring of the code, updated physics parametrization and bug fixes. These changes were not considered 'climate changing' in standalone mode, and most changes affected non-default physics. CICE6 has been shown to improve the results of Arctic sea ice compared to CICE5 and CICE4 in the standalone mode (Wang et al., 2020). However, some code changes, such as the ones related to salinity and fresh water flux calculations, might affect ocean coupled simulations.

Line 71: Naughten et al. 2018, which the baseline MetROMS-Iceshelf simulation seems to follow, used elastic-anisotropic-plastic rheology rather than EVP. What was the rationale for changing this and what was the impact? Are there any other major changes from the original model? I note that the total ice shelf basal mass loss was higher in Naughten et al. than any of the four simulations presented here, and I'm curious as to why.

Based on a visual inspection of a short test simulation, we concluded that the EVP rheology more

[Figure]

Figure 4: Fig. B7 in the manuscript:Figure of the bounding boxes used for the ice shelves for the comparison in Table 2

.

realistically reproduced sea ice dynamics than the EAP rheology. Moreover, EVP is the CICE6 default rheology.

When starting this work, we acquired a version of the circumpolar setup for MetROMS-Iceshelf from the University of Tasmania. We are not sure if this is exactly the same version as that of Naughten et al. (2018b). To our knowledge, other significant differences do not exist. However, minor differences between the MetROMS-Iceshelf and our simulations include different values of the CICE parameter "the e-folding scale of ridged ice", for which we use the default value of `mu_rdg=3`, instead of `mu_rdg=5` as in the MetROMS-Iceshelf. A short MetROMS-Iceshelf spinup simulation, which was not included in Naughten et al. (2018b), had a sponge layer at the northern boundary, that we did not have.

We agree that looking into the reasons of the differences in the basal mass loss would be interesting. We have, however, not looked into why Naughten et al. (2018b) has a higher basal mass loss, and we consider such assessment outside the scope of this work.

Figure 1b: The two different blue shadings are hard to distinguish, could you use two more different colours?

Thank you for the suggestion. We have updated the figure as shown here (Fig. 5).

Line 128: As with CICE6, the improvements made in ERA5 are summarised as "development in model physics", which could mean any number of things. Which specific processes have been improved?

The ERA reanalyses uses the 4D-Var data assimilation system IFS (Integrated Forecasting System). The model version of ERA-Interim is 31r2 from 2006 while ERA5 uses 41r2 from 2016, so the modelling system has seen multiple improvements over ten years. In the documentation of physical processes of IFS Cycle 41r2 (ECMWF, 2016) almost all subsections report changes since 2006. For example, in the Radiation scheme, the cloud scheme and the surface parameterisation have changed. Additionally, the

[Figure]

Figure 5: Colours have been updated in Fig. 1b of the manuscript. Labels A, B, and C indicate the Ross, Filchner-Ronne, and Amery Ice Shelves, respectively.

horizontal resolution between ERA-Interim and ERA5 has increased from around 80 km to 31 km, the data assimilation has been improved remarkably, and ERA5 includes a 10-member ensemble providing uncertainty estimates (Hans Hersbach et al., 2019).

Line 137–140 of the manuscript have been updated to summarise this information:

ERA5 is a considerable upgrade from ERAI with a finer horizontal resolution of 31 km compared to the ERAI 80 km. The underlying IFC (Integrated Forecasting System) modelling system has seen multiple improvements over ten years, with development in model physics, numerics, and data assimilation (Hersbach and de Rosnay, 2018).

Line 134: Note that this salinity restoration is only at the surface.

We have added a mention that this is the surface to make it clearer in the manuscript on line 149–150:

The same salinity restoration scheme is applied to the surface of ROMS in all the model runs, as previously used in MetROMS-Iceshelf by Naughten et al. (2018b), where it was used to prevent deep convection in the Weddell Sea.

Line 139: Why is the salinity restoring necessary? It's an unfortunate limitation on the model and the sort of experiments for which it is suitable. Given the updates to ERA and CICE, have you tried switching it off?

Thank you for the question. According to Naughten et al. (2018b) the salinity restoring was needed to prevent deep convection in the Weddell Sea in MetROMS-Iceshelf, so we decided to keep it on for all model simulations. We have not tried switching off the salinity restoration scheme. Future plans include trying to switch it off or changing it for MetROMS-UHel to take into account the sea ice, but this work is outside of the scope of the current manuscript.

This has been clarified in lines 149–150 cited above.

Line 139: ABW should be AABW.

Corrected, thank you.

Line 141: Why is there zero zonal velocity at the northern boundary? This seems equivalent to a no-slip condition, i.e. treating 30S as a solid wall. I wonder if this is contributing to slowing down the ACC, as noted later. Or, is it sufficiently clear of the northern front of the ACC?

Here we just tried to follow what was made in the previous MetROMS-Iceshelf setup in the input files we acquired. Naughten et al. (2017) states the choice was made to "prevent waveguide artifacts". We have not performed tests about this, and chose to follow the original configuration.

Line 185: Are the biases in spring (too much ice offshore and too little close to the coast) linked to the strong sea ice drift shown later (i.e. excessive export of ice)? Or are they the beginnings of the summertime low bias, driven by thermodynamic melting?

We think that the excessive export plays a major role, but the effect of thermodynamics also plays a secondary role. The effects of dynamic and thermodynamic processes are often interlinked and hard to separate. However, looking at the spring sea ice concentration in Fig. 2 of the manuscript, and the sea ice drift from winter in Fig. 8) (Fig. 8 in the revised manuscript), we can see that the drift usually points to the areas of overestimated sea ice concentration. Additionally, from Fig. 6 we can see that, in spring, the pack ice area is underestimated, while the marginal ice zone is overestimated. The sea ice is therefore too spread out, probably due to the excess transport.

We have added a short mention on this link to the drift chapter Sect. 4.1.5, lines 397–398:

The observed large speeds could explain the biases seen in the sea ice concentration in spring (OND), discussed in Sect. 4.1.1.

Line 198: Why might the Pacific and Indian Ocean sectors show the largest underestimation? They are quite a different regime to the other sectors, oceanographically.

We think this might have to do with the ocean hydrography, especially in the Pacific sector, where the warm CDW comes further south in the model runs than where we see it in EN4, and for the bottom of the shelf in the Schmidtko et al. (2014) comparison in Fig. B6.

The updated sentence in Lines 213–215 now reads:

Throughout the year, the largest underestimations happen in the Pacific and Indian Ocean sectors, where especially in the Pacific sector the shelf water seems to be too warm (Fig. B6), and the warm CDW seems to get too far south (not shown).

Figure 4: It would be interesting to fit linear trends to the observations and models, perhaps piecewise breaking around 2014.

Thank you for the suggestion. Fig. 7 shows the time series of sea ice area including linear trends. However, in order to avoid adding a crowded figure to the manuscript, we have decided to include Table 2 (as Table B1 in the revised manuscript) with the means, trends and correlation coefficients (as suggested by Referee 2). Corresponding text referring to the figure has been updated to discuss these values in lines 241–253.

Line 310: Do the CICE6 simulations generally have younger sea ice (there should be an age tracer to analyse)? This would make sense, together with less ridging, thinner coastal ice, stronger export, and possibly stronger sea ice formation (see my general comment).

Unfortunately, the ice age tracer was not saved as an output in the model simulations, so we are not able to answer this question.

Figure 7: It's really hard to see the differences between simulations. Could you plot anomalies of the ice speed without the vectors, perhaps in supplementary?

We have updated the figure (Fig. 8) with differences between simulations as Fig. 8 in the revised manuscript, and added a reference to it in the text.

Line 338: I don't understand what is meant by "simple ocean boundary". Do you mean the sea ice-ocean interface? What is simple about it?

[Figure]

Figure 6: Seasonal cycles of a) sea ice extent, b) sea ice area, c) back ice area and d) marginal ice zone

Here we refer to the boundary between land and ocean, not ice and ocean. As our resolution at the coast on the continental shelf is 8–10 km resolution, with some smoothing of the topography and land/sea mask, the model lacks features like grounded icebergs and other coastal features that in real life would anchor the sea ice, decreasing the amount of drift. We have striven to make this clearer in the text (Lines 399–404):

Furthermore, the relatively low resolution of the ocean-land boundary at the coast and the lack of grounded icebergs could also contribute to the overestimation. A higher resolution ocean-land boundary including icebergs would, potentially, cause slower average motion of the sea ice and longer surviving ice in summer (Naughten et al., 2018b). The low resolution ocean-land boundary might also be a reason for the underestimation of the sea ice, especially in the summer, as it has been shown that sea ice transport is an important process during melt season (Goosse et al., 2023).

Line 358: How appropriate is a comparison to EN4 on the continental shelf (let alone the missing ice shelf cavities)? Does it include enough reliable observations on the shelf?

We acknowledge that the EN4 is based on sparse data in the Antarctic ocean. However, we decided that sparse data to compare to is better than no data at all. The average temperature and salinity error standard deviations and observational weights can be seen in Fig. 9. The lack of sea ice cavities in EN4

[Figure]

Figure 7: Timeseries over 1992–2022 of (a) February and (b) September monthly average sea ice area [$10^6$ km$^2$], with 1992–2014 and 2014–2018 trend lines for the four different model runs (blue, green, yellow, red) and Sea Ice Index (grey), as well as NSIDC CDR sea ice concentration (black). The mean and slope of the trend lines can be found in Table 2

has been taken into account, for example, in Fig. 10 of the manuscript, where the MetROMS zonal means are calculated on the same grid, leaving out the ice shelf cavities.

We have made this clearer in lines 425–427:

It is important to keep in mind, when interpreting the results, that observational data from the Southern Ocean are sparse resulting significant uncertainties in the EN4 data.

Figure 9: This figure is uncomfortably similar to Figure 4 of Naughten et al. 2018, down to the placement of the labels and the nonlinear colour bar. Was the same code used? If so, no attribution is given. I am not sure of the journal's policies on this.

The code is based on the code for Fig. 4 of (Naughten et al., 2018a), but is not identical. Attribution to this has been given in the code in Äijälä and Nie (2024), and we have added a clearer mention of attribution to the metadata of the Zenodo publication as well as to the acknowledgments of the manuscript.

Line 370: The freshening of HSSW in CICE6 is concerning. Hopefully some further tuning of the new sea ice model parameters could alleviate this. Why do you think it occurred? Presumably there's been a decrease in ice formation and/or a local increase in melting — either way, what could cause this? I don't understand how this agrees with the other sea ice variables suggesting higher formation (see my above comment on line 310) or deeper mixed layers in the key formation regions (analysed later).

We agree that this is concerning and hope that tuning the sea ice parameters or modifying the surface salinity restoration will alleviate this in future efforts.

We made a shorter test run with C5E5 and C6E5, where we saved as output the CICE and ROMS parameters for freshwater flux and salinity flux, which are passed to ROMS, and used in ROMS to calculate the salinity flux. We looked into the first year of the spinup, where the sea ice have had the

[Figure]

Figure 8: Fig. 8 in the manuscript: Mean sea ice velocities during winter (JAS) over the period 1992–2018, overlaid on the speed from (a) OSI-455, (b) NSIDC ice drift and (c–f) model outputs, and differences between sea ice velocities between model runs (g–i). Only grid cells with climatological winter (JAS) sea ice concentration larger than 15 % are plotted.

| Variable | 1992-2014 Mean | 1992-2014 Slope | 2014-2018 Mean | 2014-2018 Slope | 1992-2018 correlation to NSIDC |
|---|---|---|---|---|---|
| **February** | | | | | |
| C5EI | 0.90 | 0.000 | 0.88 | -0.121 | 0.490 |
| C5E5 | 1.12 | 0.004 | 1.17 | -0.148 | 0.683 |
| C6EI | 0.86 | -0.001 | 0.78 | -0.136 | 0.471 |
| C6E5 | 1.18 | 0.001 | 1.12 | -0.193 | 0.655 |
| Ice Index | 2.05 | 0.025 | 2.06 | -0.294 | 0.980 |
| NSIDC | 2.23 | 0.021 | 2.17 | -0.316 | - |
| **September** | | | | | |
| C5EI | 15.02 | 0.017 | 14.44 | -0.435 | 0.801 |
| C5E5 | 15.38 | 0.004 | 14.73 | -0.442 | 0.804 |
| C6EI | 15.38 | 0.033 | 14.80 | -0.532 | 0.840 |
| C6E5 | 15.70 | 0.018 | 15.07 | -0.566 | 0.868 |
| Ice Index | 14.74 | 0.036 | 14.44 | -0.363 | 0.960 |
| NSIDC | 16.56 | 0.035 | 16.18 | -0.384 | - |

Table 2: Table B1 in the manuscript: Mean and slope values for timeseries for sea ice area in February and September (Fig. 4): 1992-2014 and 2014-2018 as well as the Pearson correlation calculated against NSIDC for 1992-2018

least time to evolve differently in the models.

The average salinity flux in ROMS for the first year of the spinup was positive at much of the coast, and mostly slightly negative otherwise. The fluxes are smaller in CICE6 than in CICE5, except close to the maximum ice edge and in parts of the Weddell Sea. The difference is clearly largest at the coast.

In CICE, two fluxes from ice to ocean are passed to ROMS, the freshwater flux (fresh_ai) and the salinity flux (fsalt_ai). Both of these have very similar spatial patterns. When ice is formed, both water and salt are removed from the ocean and the fluxes are negative, and when sea ice melts, freshwater and salt are released to the ocean and the fluxes are positive. These fluxes are on average larger in CICE6 than in CICE5, especially at the coast, similar to the ROMS salinity flux, so that when the ROMS CICE6-CICE5 is negative the CICE salt and freshwater flux CICE6-CICE5 is positive (Fig. 10a).

We looked into the difference in coastal ice growth between CICE6 and CICE5 to see if the difference in the fluxes could be explained by ice growth differences, but as can be seen in Fig. 10b, the difference in ice growth is not consistently positive or negative at the coast. CICE6 has on average a larger ice growth, but there are also areas where CICE5 ice growth is larger, while its fluxes are consistently larger.

This pattern of difference in fluxes correlates much better with the overall ice growth, so that the flux difference increases as ice grows (Fig. 10). This indicates that the flux difference related to the amount of salinity/freshwater flux due to the ice growth, not due to the difference in ice growth between the CICE models.

The text in Lines 482–487 has now been made clearer:

This bias can be attributed to the salt flux from CICE to ROMS which is, on average, smaller in CICE6 than in CICE5. The largest differences can be found at the coast and when the salinity flux is positive, i.e. from the ice towards the ocean, where the ice growth is largest (Fig. 7). The difference does not correlate with the change in ice growth, and seems to be an effect of a change in the flux calculation in CICE (not shown). Because the ice formation rate does not seem to be the cause for the freshening problem, it is likely related to how the CICE's salinity and freshwater fluxes are handled and converted to ROMS salinity flux.

Line 384: The deep waters offshore will basically just be initial conditions so early in the simulation; this should be made more explicit in the note of caution.

We have made this clearer in the text (Lines 458–459):

Deep waters take a long time to spinup due to longer residence times, as discussed in Sect. 3.1, and will therefore be the same or very similar to the initial conditions, and should be interpreted with caution.

[Figure]

Figure 9: EN4 1992–2018 and depth level averaged uncertainty. Top left shows temperature error standard deviation [K], top right shows temperature observation weights, bottom left shows salinity error standard deviation and bottom right shows salinity observation weights.

Line 398: How is the model surface more saline than EN4 when surface salinity is nudged to observations? Is the nudging really weak, or do the two datasets (WOA and EN4) disagree?

A bit of both. The nudging is quite weak, but the nudging calculated from WOA is also, on average, more saline south of 60° S (Fig. 11). In C5EI the salinity flux from nudging is mostly negative, except in the northern parts of the domain, while in C6E5 we have areas in the south with nudging being on average positive, but there, some of the surface is less saline than EN4, not more saline.

Line 407: A better way to mitigate the freshening problem would be to tune up the CICE6 sea ice formation rates. Salinity restoration should be a last resort, especially on the continental shelves.

This is a good point. However, the ice formation rates do not seem to be the cause for the freshening problem (see answer for comment for Line 370). Nevertheless, it would be good to find a way to handle the effect that the change in salinity fluxes have, to see if something in the coupling or the CICE settings needs to be changed here. We have not really looked into how the conversion of CICE's freshwater and salinity flux is handled when converted to the ROMS's salinity flux.

Some updates have been made to the text in this regard, as mentioned above. additionally lines 487–490 reads:

Looking into this would be a good place to start when tuning the model in the future. The salinity restoration scheme could also be updated. The current salinity restoration scheme is only applied when the ocean is deeper than 1500 m (Sect. 3.2) and does not take sea ice into account. Testing different salinity restoration schemes is beyond the scope of this work.

Figure 13: Presumably the colour scale is saturated, as observed melt rates in the Amundsen and Bellingshausen sectors are much higher than 4 m/y. The colour scale should indicate this with triangle caps.

[Figure]

Figure 10: (a) One-year average of fresh_ai difference between C6E5 and C5E5, (b) one-year average of difference of sea ice growth (snoice+frazil+congel) between C6E5 and C5E5 and (c) one-year average sea ice growth in C6E5.

[Figure]

Figure 11: Difference between the average of the sea surface salinity nudging climatology based on WOD and EN4 for the period 1992–2018

Yes, you are correct; the colour scale is saturated. Thanks for pointing this out. We have fixed the colour scale in Fig. 13 and in other figures which had the same problem.

Line 464: The wording implies that the biases in the BellAm sector are secondary, but I suspect they are the main driver of the low total mass loss. The cavities are small, so hard to see in Fig. 13, but their mass loss is huge. This would be easier to see in the bar chart I suggested.

You are correct, from the added table (Table 1) we see that the largest negative bias is in the Amundsen and Bellinghausen Seas together with the Australian sector. Of these, the Amundsen Sea has the largest underestimation compared to Susheel Adusumilli et al. (2020), both in absolute and percentage terms. We have mentioned this in the reworked text on lines 551–557:

The largest underestimations, compared to both observational datasets, occur in the Australian region as well as the Amundsen and Bellingshausen Sea regions (Table 2). In these regions, all the models show values far below the range given by Rignot et al. (2013), except in the Wilkins Ice Shelf in the Bellingshausen Sea, where uncertainty is large. In the BellAm sector, this is likely due to the warm shelf area being too cold in the model runs (Fig. B6b), while this is not the case for the Australian region, which is warmer than observations in the model runs (Fig. B6Bb). Naughten et al. (2018b) speculates that the underestimation of the Australian region is due to a lack of HSSW.

Line 367: Does the Amundsen Sea have the highest melt increase in percentage terms, or only absolute terms? An extra 2 m/y is not a big change for warm cavities like these, compared to cold cavities with much lower initial melt rates.

The Amundsen Sea region has the highest increase in melt both in absolute and percentage terms. This is clearer in the new Table 1.

Line 471: It looks like there is also a loss of refreezing beneath the Ross and Filchner-Ronne Ice Shelves. This would make sense if fresher HSSW is dampening the magnitude of the ice pump.

There seems to be some loss of refreezing in the middle parts of the Filchner-Ronne Ice Shelf, but it seems to be increasing in the western part of the Ronne Ice Shelf. Under the Ross Ice Shelf similar

patterns are harder to see. When calculating separately the mass change from grid cells with positive and negative melt rates (Table 3), we see that in the Filchner-Ronne Ice Shelf the negative sum actually increases, meaning there is more refreezing in the long-term average in CICE6. Under the Ross Ice Shelf we see a small decrease, but the change is small.

|  | FRIS massloss | FRIS refreeze | Ross massloss | Ross refreeze |
|---|---|---|---|---|
| C5EI | 75.73 | -10.94 | 64.60 | -2.74 |
| C5E5 | 77.84 | -11.12 | 58.11 | -2.83 |
| C6EI | 73.13 | -11.33 | 62.27 | -2.60 |
| C6E5 | 75.20 | -11.67 | 54.54 | -2.71 |

Table 3: Average mass loss over the period 1996–2018 for the Filchner-Ronne Ice Shelf (FRIS) and Ross Ice Shelf separated into positive (mass loss) and negative (refreeze) sums [Gt/yr]

Line 473: I disagree that tides and spatial resolution are the main drivers of low simulated melt rates. I strongly suspect the main driver is the cold bias in the Amundsen Sea (Figure A4), and it's arguable whether or not spatial resolution contributes to this. It's definitely possible to simulate enough CDW transport onshore in a quarter-degree C-grid model; see for example Mathiot et al. 2017 (doi:10.5194/gmd-10-2849-2017). The Nakayama study about topographic flow used a model equivalent to an Arakawa A-grid, whereas a C-grid as used for ROMS is a lot more forgiving. But even if the topography is adequately resolved, the heat from CDW could be wiped out if there is convection on the Amundsen Sea continental shelf, as sometimes happens with ERA forcing even at much higher resolution (eg Bett et al. 2020, doi:10.1029/2020JC016305). Tides could matter for the Filchner-Ronne Ice Shelf, but this is not a big contributor to the Antarctic total.

We agree with the Referee. The cold bias in the Amundsen Sea seems to be indeed one of the main drivers of the low melt rates, with ice shelves in the Amundsen Sea, Bellinghausen Sea and the Antarctic region being the areas with strongest underestimation compared to the observations (Table 1). Sect. 4.3 of the manuscript have been rewritten while adding the table to the manuscript as Table 2.

Line 523: It's a bit of a red flag that ChatGPT was used to write some of the code. Was this just for things like figure layout, or did it actually handle the data analysis? If so, was there sufficient human oversight to make sure it didn't introduce any bugs? I am not sure of the journal's policies on this matter.

ChatGPT was used to generate code snippet from pseudocode like descriptions for some of the scripts. It did not by itself handle any data analysis and all code acquired from it was checked and tested by the corresponding author.

The journal states in its submission guidelines that "Should you have used AI tools to generate (parts of) your manuscript, please describe the usage either in the Methods section or the Acknowledgments", and this is what we have striven to do when stating it was used for some code generation.

**1 Anonymous Referee #2**

**General comments:**

In this study, the authors describe an updated version of the MetROMS ocean/sea ice/ice shelf regional Southern Ocean circulation model. Four simulations were run: a base case with the original setup, a simulation where just the atmospheric forcing is updated from the ERA-Interim reanalysis to ERA5, a simulation where just the sea ice model is updated from CICE5 to CICE6, and a simulation where both the atmosphere and sea ice model are updated. The hope is that these changes improved the model simulation of Antarctic sea ice and upper ocean hydrography. Results from the different simulations were compared against each other and observations for sea ice concentration, sea ice area, sea ice edge, sea ice volume (although this is not compared to observations), sea ice drift, ocean temperature and salinity, surface mixed layer depth, and ice shelf basal melt.

I thought the manuscript was clear and easy to understand and generally met its objective of explaining how the upgrades impact the model simulation of different aspects of the Southern Ocean. This setup (ROMS ocean model and CICE6 sea ice model) is a good tool and I'm glad the authors have updated it and made it available to everyone and are using GMD to tell the community what they have done. I felt the specific quantities being compared did a good job of showing the general impact of the upgrades for simulating the physics of the Southern Ocean (although I do have specific suggested additions below).

However, there is not very much on what specific changes in either the forcing or the ice model code led to these differences in the model simulation (there is a little in lines 235–240 about biases in cloud cover and temperatures impacting summer sea ice). I certainly understand how a thorough examination of the causes is well beyond the scope of what the authors are trying to do here. However, are there any large changes between the forcing or ice code versions that are likely to lead to any of the changes seen in the results? For example, does the significant increase in the resolution from ERAI to ERA5 (thus better resolving steep coastal orography) lead to generally stronger winds along the coast, thus impacting ice motion near the coast (e.g. lines 340-341) and perhaps ice production and mixed layer depths over the continental shelf?

We thank the Referee #2 for their comments. We appreciate the Reviewer prompting us to examine and discuss the causes of the differences in the simulations, as we think this can significantly improve the manuscript. Below, we address these inquiries in the best way possible.

One of the major differences between ERA5 and ERA-Interim is the changes in the resolution. While ERA-Interim has a horizontal resolution of 80 km, ERA5 represents a significant improvement with its 31 km resolution. This improved resolution is particularly relevant to resolve small-scale processes at the coast (Fig. 12). For the sea ice, the difference in wind seems to play a relevant role, with slower winds at the open ocean, and stronger winds close to the coast to around 65° S, and then again areas of slower winds at the coast, probably due to a change of topography. These patterns are well correlated to an increased sea ice drift close to the coast, and a decrease of sea ice formation at the coast. We can also see clear differences in most of the other forcing fields, but they are not as easy to connect to the changes seen in the model runs without a deeper analysis. We have added Fig. 12 to the Appendix as Figure B1. Further discussion about these aspects has been included throughout the text, when answering to specific comments of both Referees. Specifically, a sentence in Lines 144–146 in Sect. 3.2:

The largest differences between the ERA5 and ERAI forcing (Fig. B1) are observed mainly at the coast, where the increased resolution helps resolving small-scale processes, for example for the wind, that see large changes at the coast.

The documentation of the sea ice code indicates that no changes to the basic physics should be 'climate changing' in standalone mode, but for example, a change in salinity flux calculation might affect coupled models (see answer to Referee 1 specific comment Line 68). This change in salinity fluxes seems to be behind the freshening in the ocean. Other than that, it is not easy to pinpoint what the reason behind the seen changes are as all processes affect each others in the coupled system.

We think the Referee is correct with their thoughts that stronger wind along the coast might be a reason to the increased mixed layer, and have added the following text to the MLD Sect. 4.2.2 in Lines 508–509:

The observed pattern of MLD increase on the continental shelf and decrease in the open ocean can probably also be linked to changes in wind fields between ERAI and ERA5 (Fig. B1e,f).

I also felt that there could have been more to tie the presented individual changes to each other. For example, for the basal melt there is a discussion of what may cause the systemic underestimation for all the models compared to observations, but there is no discussion of what differences in the modeled hydrography between the simulations would lead to the shown differences in basal melt between the different simulations. Changing the ice model generally leads to greater ice shelf basal melt (especially over the Amundsen, Fig. 13d), which could be related to the temperatures at depth over the continental shelf generally being warmer (Fig. 10e), and the temperature difference between C6EI and C5EI might even be greater over the deep Amundsen shelf (hard to tell as there is no plan view of the temperature difference) as the winter mixed layer depths are significantly shallower there with the ice model change (Fig. 12f). Updating the winds leads to generally lower ice shelf basal melt, which can be related to lower

[Figure]

Figure 12: Figure B1 of the manuscript Differences between ERA5 and ERAI forcing fields calculated from 1992-2018 average. For Uwind and Vwind the plots (e,f) shows the change of the strength of the wind component in it's direction calculated as $wind\_diff = (wind\_era5/abs(wind\_era5)) * (wind\_era5 - wind\_eraI)$, and the gray lines indicates the areas where the direction of the wind vector flips into the opposite direction.

temperatures at depth (Fig. 10d) over the continental shelf.

Thank you for the comment. We have added more discussion throughout the text to connect the changes to each other, many of these changes are cited in other answers. However, we have not done a in-depth analysis on the changes under the ice-shelves as that would be out of the scope of this manuscript, but you are correct, the temperature under the ice shelves in the Amundsen Sea as well as under the Amery Ice Shelf increase, leading to the increased basal melt. We have added this discussion and more to the melt rate section (Sect. 4.3).

I have some other specific comments and suggestions below, but most of these are minor and should be easily dealt with by the authors. I think this upgrade is important, but I also think this description of the impacts needs a bit of work before it will be fully useful for the intended audience.

We thank again the reviewer for these suggestions. Please see our point-by-point response below addressing the reviewer's comments.

**Specific comments:**

Lines 39-41: Is it worth adding something here that while satellite based estimates of thickness are now becoming available in the Arctic, these efforts are not nearly as well advanced in the Antarctic since it is harder to estimate freeboard? I know this is briefly mentioned at the beginning of section 4.1.3, but since this is another example of a difference between southern and northern sea ice, should it be mentioned here?

Thank you for the comment. We have added a mention in Lines 37–39 about the disparity between Arctic and Antarctic satellite-based estimates:

And while satellite-based estimates of sea ice properties, such as sea ice thickness, are becoming available in the Arctic, these efforts are not nearly as well advanced in the Antarctic.

A sentence on the complexity of snow and sea ice conditions was also included in Lines 44–46:

The Southern Ocean receives more snowfall than anywhere else in the world (Lawrence et al., 2024), and the snow distribution on the sea ice is more complex than in the Arctic due to warmer winter temperatures and frequent snow flooding (Willatt et al., 2010).

And the difficulty of estimating freeboard in satellite measurements was included at the beginning of Section 4.1.3 such that Lines 300-302 read as follows:

Sea ice thickness observations in the Southern Ocean are scarce and have large uncertainties (Holland et al., 2014; Uotila et al., 2019; Xu et al., 2021), largely due to the complexity in the snow cover, which makes it challenging to measure freeboard of the sea ice from satellites and affecting satellite-based estimates of ice thickness (Giles et al., 2008).

Lines 70-71: I think it would be helpful to any potential users to explicitly list somewhere (maybe a supplemental table?) the instances in which CICE default values are not being used.

Thank you for your comment. The CICE6 input file, ice.in has been provided as part of the model code in Äijälä and Uotila (2024) in the "example_input" folder. This file shows exactly what input parameters were used, and we will not add a table of the parameters to the manuscript, as the manuscript is alreaady quite long.

Lines 86-87: Is the freeze-melt potential from ROMS integrated over the top 5 m of the ocean model for both versions of CICE as in Naughten et al. 2017? If so, since there could be multiple ROMS layers in the top 5 m, is supercooling in the layers below the surface layer also handled as described in Naughten et al. 2017?

Yes, the freeze-melt potential is handled in the same way. The following text has been added at lines 97–98 to clarify:

The freeze–melt potential, following (Naughten et al., 2017), is integrated over the top 5 m of the ocean model.

Line 123: I really feel the ACC transport for the different experiments should be given.

Thank you for the comment, this was also a concern raised by Reviewer #1. Following this advice, we have added a section about DPT in Appendix A of the revised manuscript, where a short discussion is given. Appendix A includes Fig. 1 (Fig. A1 in the revised manuscript), which is now referenced in Line 134.

Section 3.2: I know it is mentioned in section 4.3, but I think it should be explicitly mentioned somewhere in this section that there is no tidal forcing.

We added a mention about the lack of tidal forcing in Section 3.2 (Lines 147–148):

The model does not have tidal forcing as tides are not accounted for.

Lines 150-152: Is there also a sponge region near the northern boundary with increased viscosity and diffusivity as in Naughten et al. 2017?

We did not use a sponge region near the northern boundary.

Line 162: Suggest changing "increases the concentration" to "generally increases the concentration" since there are areas where the ice concentration decreases with the updates.

We followed this suggestion, thank you.

Lines 221-222 and Figure 4: I think it would be helpful to the reader to include a numerical comparison (correlation or skill score maybe?) between each of the different modeled sea ice area time series and the observations.

We have added a table to the Appendix B. Table B1 includes correlations, means and slopes following this suggestion and that of Referee #1.
References to this table with some numerical comparisons have been added through the text referring to the figure in lines 241–253.

Lines 307-316: One other metric that I think would help in determining why there are increases in concentration but decreases in volume near the coast when updating the ice code is the thermodynamic sea ice production, which could be compared to observations (e.g. Nihashi and Ohshima, 2015; Nakata et al., 2019) and help determine if volume decreases near the coast are due more to decreased production or changes in ice movement.

We have added a figure on growth and melt following the suggestion from the general comment 3 of Referee #1 (Fig. 7), but we consider that making a comparison between our model runs and the suggested polynyas ice growth publication is out of the scope of this paper, as it would require considerable work to be addressed properly.
We have however mentioned this phenomena in rows 360-361 in the new Section 4.1.4:

This supports the argument, presented in Sect. 4.1.3, that the decrease in volume is connected to the change in ridged and level ice area fractions (Fig. B4).

Figure 7 caption: Is the "climatological sea ice concentration" the climatology during winter (JAS) or over the entire year?

It is the climatology during winter (JAS). The legend in Fig. 7, Fig. 8 in the revised manuscript, has been updated (Fig. 8).

Line 346: "Robust" seems like a strong word for the level of increases shown in Fig. 8 (e.g. from 0.457 to 0.466 in the Indian Ocean sector or 0.483 to 0.493 in the BellAm sector) between C5EI and C5E5.

Thank you for your comment. We have remove the word "robust" from the sentence now in Line 411.

Figure 9: I believe the ROMS model output is not absolute salinity, so my guess is that the authors converted the output to absolute salinity. Apologies if I missed it, but should that be explicitly mentioned

You are correct, the model output is given as practical salinity and has been converted to absolute salinity in the analysis using the TEOS-10 standard. This information has been included in the captions of Fig. 9 and Fig. 10 of the manuscript.

Lines 405-407: This is more reason why I think it would be helpful to show the sea ice production from the different runs.

The change in salinity flux is not due to a change in sea ice growth, but seems to be due to a change in CICE instead. See answer to Referee # 1. comment for line Line 370.

Lines 428-429: It is hard to tell from the figures here that there is "a small improvement of the summer MLD". Can the authors make a table of overall RMSE between the MLD of each of the different runs and Sallée (or add a column to Fig. 11 with C6E5 – Sallée)?

We have added the suggested column to the Fig. 11, now Fig. 12.

Line 441: Suggest adding "excess" between "possible" and "deep water formation" since there should be deep water formation in these areas even if the MLDs there were perfect.

The word "excess" has been added as suggested.

Line 457: The 578 Gt/yr for C5EI is a fair bit lower than the 642 Gt/yr in Naughten et al. 2018b. Do the authors have ideas on what is causing the difference, other than the different (1996-2018 vs. 2002-2016) time periods?

The C5EI is indeed lower than that in Naughten et al. (2018b), but we have not looked into this difference in detail. We do not have the exact setup used in Naughten et al. (2018b), nor the output data for that run, so further analyzing this difference, or the reasons behind them is challenging and outside the scope of this work. However, we notice that the difference is not only due to the use of different time periods, as calculating the C5EI mass loss in 2002-2016 gives us a mean of 583Gt/yr.

Looking at the mass loss comparison between different regions made at the request of Referee #1 (Table 1), as well as the shelfwise table it is based on, and comparing it to Table 1 in Naughten et al. (2018b), the biggest absolute differences seem to be in Amundsen Sea where Naughten's melt rate is 125.1 Gt/yr and our C5EI run is 83.7 Gt/yr. Here, all our mass losses for the separate ice shelves are smaller. We also see clearly smaller values in our run for Bellinghausen Sea and eastern Weddell region, while our run has a clearly higher mass loss in the Filchner-Ronne ice shelf. We have not looked into the reasons behind these differences, and feel that to be out of the scope of the manuscript.

An additional note: Due to small corrections in the code, the value of 578 Gt/yr should actually be 577 Gt/yr, due to rounding differences.

Line 461: Suggest adding "model" between "The" and "refreezing" to differentiate the melting in this sentence from the preceding one.

The word "model" has been added as suggested, thank you.

Lines 472-485: This discussion on why all the models have too little melting is interesting, but most of this is already discussed in Naughten et al., 2018b (their sections 4.3 and 4.3.6), which should be referenced here. Also, while it's certainly important to mention this large difference, since this paper is mostly about the upgrades, shouldn't there be something on why there is a difference in the melting with the atmospheric forcing and ice model updates?

Thank you for the comment. We have rewritten most of the melt rate section (Sect. 4.3), added clearer citations to Naughten et al. (2018b), and added discussion on reasons for the changes.

**Technical corrections:**

Lines 6-7: I don't think the sentence beginning "Both CICE sea ice models..." is necessary.

Thank you for your comment. This sentence has been modified to:

'The two versions of MetROMS evaluated in this study use a version of the regional ROMS ocean model including ice shelf cavities.

Line 8: "increase" should be "increases".

This has been fixed.

Line 10: Should "ocean mix layer" be "ocean mixed layer"?

Thank you for pointing this out, we have fixed this.

Line 96: Should the Filchner-Ronne, Amery, and Ross ice shelves be labelled somewhere (Fig 1b)?

We have added labels to Fig. 1b of the manuscript, as seen in Fig. 5 of this document.

Line 208: "being C6E5" should be "with C6E5 being".

Thank you for pointing this out, the sentence has been corrected.

Figure 6 caption: The units should be "([model volume]/[C5EI volume])".

Thank you for pointing out the error. The caption has been corrected.

Line 312: Typo, "A3)a-d" should be "A3a-d".

This has been fixed.

Line 358: Should the ";" be " and"?

Thank you for pointing this out, this has been corrected.

Line 370: "them clearly fresher" should be "them are clearly fresher" or "them clearly are fresher".

Thank you, this has been corrected.

Line 475: The "." After "basal melting" is not necessary.

The full stop has been removed.

[revised manuscript text omitted]

---

## Author Response (AR2)

**Response to Referee Comments**

May 7, 2025

The document is color coded in the following way:
Referee comments are in **blue**.
Our answers are in **black**.
Citations from the revised manuscript are in **red**.

**Anonymous Referee #1**

The authors have mostly done a good job at responding to my review. I now have one major comment, and a few more minor ones:

We thank the reviewer for their thorough review, and address the comments below:

**Major comment:**

I was surprised to see the authors' analysis that the CICE6 flux calculation itself was the cause of the coastal freshening in ROMS, not changes in ice growth or melt. Later in their response the authors say they have not looked into the MetROMS coupling calculations to see if they should be updated. I think this should be an urgent priority, as something like a subtle unit change (e.g. scaled by ice concentration or not) could have a large effect over time. Best practice is to review the coupling any time there is a major version change in a component, and I recommend this is checked before the manuscript goes any further.

The earlier answer might have been misleading. When we said that *"We have not really looked into how the conversion of CICE's freshwater and salinity flux is handled when converted to the ROMS's salinity flux."*, we meant that we have not tried to re-tune this, but we have checked that nothing should have changed regarding the CICE outputs according to CICE documentation.

If the coupling is correct, a brief discussion of the implications would be warranted: given that the community has spent many years tuning ocean models coupled to CICE5, some re-tuning will be necessary in the future to deal with the new flux behaviour in CICE6, and of course this is not trivial.

We have re-examined our coupling code and could not find any errors there. However, upon further examination of the CICE code, we found a bug regarding the fluxes to the ocean from frazil ice formation in the Icepack submodule (version Icepack 1.3.1) used by CICE 6.3.1, released 02/2022. According to the Icepack Github issue #390 https://github.com/CICE-Consortium/Icepack/issues/390, this bug was found in May 2022 and fixed by the end of 2023.

Based on this information, we plan to address this problem in the next version of the MetROMS-UHel model. We consider that trying to resolve the problem now is out of the scope of this manuscript. CICE 6.3.1 and MetROMS-UHel 1.0 have already been used in other research, and we consider that an evaluation of the current version of the model is still valuable and useful for the community.

We have added a mention of this bug in the metadata of the released model code in zenodo:

Icepack 1.3.1 in CICE 6.3.1 has a bug when using ktherm = 2 and update_ocn_f = True, affecting the salinity flux from frazil ice formation (`https://github.com/CICE-Consortium/Icepack/issues/390`). We suggest taking this into account before using this version of the model.

And discussed this problem in the revised manuscript on rows 484 to 487:

The problem seems to originate from a bug in the Icepack 1.3.1 code (submodule of CICE 6.3.1), regarding calculations of salinity fluxes from frazil ice formation (CICE-Consortium, 2022), which is strongest at the coast. The bug has been resolved in later versions of CICE, and we plan to address this issue in the next version of MetROMS-UHel.

**Minor comments:**

Figure B5: Even in the initial state, the AABW cell is extremely weak and does not extend as deep as we believe it should (suggesting the dense water is not cascading all the way down the slope) - compare for example to Figure 2a of Zhang et al. 2016 (doi:10.1002/2016JC011790). The authors should note this initial bias in the text. I am also struggling to see how this cell is stronger in the CICE6 runs, as the authors claim - it's maybe a touch stronger in the upper ocean but if anything weaker at depth. I also don't see how the CICE6 runs experience more weakening than CICE5 in the anomaly panels - weakening would show up as positive anomalies, but (at least at depth) these seem more positive for CICE5. Maybe a different colour scale/contours, plotting in density space, or highlighting the regions of interest would help?

Thank you for this comment. We note the initially weak AABW cell in the text and have rewritten the text related to Figure B5 (lines 442–446). Essentially, we admit that there are no systematic differences in the initial AABW cells, and since they are already quite weak from the beginning, the CICE6 AABW cells can not weaken further despite the freshening of coastal waters.

Table 2: Why is everything compared to Rignot 2013 with the + and -, including Adusumilli 2020, which are also observations (and more up to date)? I think it would be more correct to compare only the modelled melt rate to the full range of both datasets (Adusumulli would need some kind of uncertainty quantification, even just a standard deviation over time would help), and consider it biased (with + or -) only if it falls outside the range of both. This takes into account uncertainty in observational estimates, and does not favour one dataset over the other without justification.

This was done as we did not have comparable error estimates for the Susheel Adusumilli et al. (2020). Upon the reviewers request, we have now instead used the melt water flux estimates from 1994–2018 Adusumilli et al. (2020) supplementary Table 1 for the 25 ice shelves used in the comparison (`https://static-content.springer.com/esm/art%3A10.1038%2Fs41561-020-0616-z/MediaObjects/41561_2020_616_MOESM1_ESM.pdf`). The values for the melt in the supplementary table differ somewhat from those acquired from the Susheel Adusumilli et al. (2020), mostly in the Australian sector, and probably due to how the edges of the shelves are defined, but not enough to affect the conclusions drawn in our study.

Table 2 and the corresponding text in Lines 530 to 557 have been updated with these new observational values.

Lines 74-79: How are the CICE6 updates not "climate changing" if they affect fluxes to the ocean, and have been shown in this and other studies to change the behaviour of the coupled ice-ocean system? Does this phrase only consider ice-atmosphere fluxes?

The CICE developers have stated that the update from CICE5 to CICE6 should not be "climate changing" in standalone mode, i.e. when not coupled to an ocean model. This statement does not take the fluxes into account. The meaning of the phrase "climate changing" is described in Roberts et al. (2018), where they state that:

*"Understanding whether or not non-BFB changes in CICE code may also alter the climate of the model can be non-trivial. By 'climate changing', we mean significant changes in sea-ice thickness, h, over a substantial fraction of the ice pack within a defined number of annual cycles. h integrates changes in sea-ice growth, melt, drift and deformation, and therefore the time series hi of ice thickness, weighted by ice concentration, documents evolution of simulated ice mass and underpins our quality control (QC) procedure (i is a time index)."*

We have made this clearer by adding the following sentence to the lines 77 to 79:

The developer defines 'climate changing' as 'significant changes in sea-ice thickness, h, over a substantial fraction of the ice pack within a defined number of annual cycles' (Roberts et al., 2018).

**Anonymous Referee #2**

**General comments**

I thank the authors for all their efforts in response to my comments, especially in adding more details in several places on how specific changes in the forcing and ice model code lead to differences in the model output (new section 4.1.4 and sections 4.2.1, and 4.3). I still think it would have been useful to compare the ice production to observational estimates, especially since the authors have now added a figure (Fig. 7) showing the model ice growth, but I can see their point that "it would require considerable work to be addressed properly" (reply to reviewers) and am happy that at least the model production is now added.

I think this manuscript does meet its objective of explaining how the upgrades impact the model simulation and this will be quite useful to some who study the Southern Ocean community. I have several new specific comments and suggestions below, but all of these are pretty minor, and I do not think this manuscript needs very much work before it is suitable for publication.

We are happy that the reviewer is satisfied with our efforts. We address all the minor specific comments below:

**Specific comments**

Line 12: Since there are areas where the modeled melt rates are greater than observations (e.g. Amery), I suggest changing "melt rates are underestimated" to something like "melt rates are generally underestimated".

Thank you for the comment, this has been changed as suggested.

Lines 37-38: Since satellite ice concentration has been available in the Antarctic for quite a while, I suggest a slight modification of "estimates of sea ice properties" to something like "estimates of sea ice properties beyond ice concentration".

We have updated these lines as suggested.

Line 76: I don't understand what is meant by 'climate changing' when referring to model code updates. Do the authors mean the changes do not impact mean climate states simulated by the model?

The 'climate changing' is citing the developers of CICE, who in Roberts et al. (2018) define the term as:
*"Understanding whether or not non-BFB changes in CICE code may also alter the climate of the model can be non-trivial. By 'climate changing', we mean significant changes in sea-ice thickness, h, over a substantial fraction of the ice pack within a defined number of annual cycles. h integrates changes in sea-ice growth, melt, drift and deformation, and therefore the time series hi of ice thickness, weighted by ice concentration, documents evolution of simulated ice mass and underpins our quality control (QC) procedure (i is a time index)."*
We have made this clearer by adding the following sentence to the Lines 77 to 79:

The developer defines 'climate changing' as 'significant changes in sea-ice thickness, h, over a substantial fraction of the ice pack within a defined number of annual cycles' (Roberts et al., 2018).

Line 134: See comments below on Appendix A, but I'm not sure "Similar decreasing trends" is accurate as the decreases here are significantly stronger than in Naughten et al. 2018b.

We have updated Lines 135 to 137 to be more accurate:

The deep ocean does not reach equilibrium and some model drift can be seen in the interior ocean, for example, in the ACC transport, measured at the Drake Passage (Appendix A), which decreases. A similar, but less negative trend of the ACC was found in the MetROMS-Iceshelf runs by Naughten et al. (2018).

Line 405: What is actually increasing in "except for an increase in both updates"?

This means that the sea ice velocities at the coast increase in both updates. Thank you for pointing out the error. This sentence now reads:

Updating from CICE5 to CICE6 or replacing ERAI forcing with ERA5 does not result in substantial changes, except for an increase of ice velocities near the coast in both updates, with the largest increase in the C6E5 run (Fig. 8).

Figure 10 caption: Suggest changing "from psu" to "from model output (psu)".

This is a good suggestion, and we have updated it as suggested.

Line 434: ISW is generated under the ice shelves, but can be found well outside the ice shelf cavities. Suggest a slight rewrite of "because it is located under the ice shelves where the high pressure lowers the freezing point" to something like "because it is generated at the base of ice shelves where the freezing point of seawater is below that at the surface due to depression of the freezing point with increased pressure".

Thank you for the suggestion. This sentence has been changed to:

This water mass can be this cold without freezing because it is generated at the base of ice shelves, where the high pressure due to depth lowers the freezing point.

Lines 439-446 and Figure B5: The text says B5 shows the streamfunction north of 60S between 3000 and 5000m, but the figure itself has axes labels for 80S to 40S and 5000m to the surface. Probably more important, the big differences in the streamfunction (assuming the x-axis label is correct) are near the model northern boundary. Because of this, and knowing that the model ACC is slowing down for some reason that may indicate significant changes in Southern Ocean interior watermasses (maybe spurious diapycnal mixing in the interior, as mentioned on lines 521-522 and Naughten et al., 2018b), I would be wary in attributing the differences in the streamfunction to differences in dense water export from the Antarctic continental shelf. I know the streamfunction was computed in direct response to a review comment about examining changes in the shelf water export, but it might be better to just examine the overturning right near Antarctica instead of over the entire model domain.

Thank you for pointing out this streamfunction issue. We replotted the streamfunctions in the updated Figure B5, where their latitudinal extent is limited between 80° S and 60° S to better see their features and differences close to the continental shelf. We have rewritten the text related to Figure B5 (lines 442–446).

Line 539: Suggest specifying that it's the melt rates that reach equilibrium in 1996 (i.e. "Our model run basal melt rates" instead of "Our model runs").

Thank you, we have updated this as suggested

Line 543: Suggest changing "decrease in the flux" to "decrease in the melt".

Updated according to suggestion

Line 556: Suggest changing "speculates that the underestimation of the Australian region is due" to something like "also underestimated melt in the Australian region and speculate that it is due".

Good suggestion. We have updated this according to the reviewers comment.

Table 2: Should the errors across the individual ice shelves in Rignot et al. be summed in quadrature to give the error for a region instead of just a simple sum (e.g. for the Australian section 198.3 +/- 20 ($sqrt(10^2 + 15^2 + 8^2 + 3^2$) instead of +/- 36 (10+15+8+3))?

Yes, they should. Thank you for pointing out the error. This has been fixed, and Table 2 and the corresponding text have also been updated following the comment from Reviewer 1.

Line 569: Suggest changing "under the Bellingshausen" to "in the Bellingshausen".

Changed as suggested

Appendix A: I know a possible cause for this trend in the ACC transport is discussed in lines 521-522, but do the authors want to add something here? Also, the decreases in all four simulations presented here are much greater than the 0.28 Sv/yr in MetROMS found in Naughten et al. 2018b. Do the authors have any speculation as to why these models have a greater decline?

We do not have any good speculation on why these models have a greater decline. We were surprised to discover it, as the two large changes that we know of between Naughten's model runs and ours are that our model runs do not have a sponge layer and the use of different rheology in CICE, with Naughten et al. (2018) using EAP and us using EVP. We do not see how these differences would result in a greater decline. There might be some other differences we are not aware of, as we do not have available the exact run setup used by Naughten, and can not duplicate her runs because of this.

We would prefer not to add anything more to the manuscript here, as a more thorough speculation would require further analysis into the reasons for the decline.

**0.1   Technical corrections**

Thank you for the thorough list of technical comments.

Line 107: "Ice Sheets" should be "Ice Shelves".

Fixed.

Line 145: I think "helps resolving" should be "helps resolve" or "helps with resolving".

Fixed.

Line 148: I think "as tides are not accounted for" can be removed from the sentence.

Fixed.

Lines 195-196: Subject verb agreement "differences . . . is".

You are correct. Replaced with "differences ... are"

Line 248: I think "Such pattern" should be "Such a pattern".

Fixed.

Line 251: Typo, "C5E6" should be "C6E5".

Fixed.

Line 375: Typo, "Fig. 3 and 3" should be "Fig. 3 and 6".

You are correct. Fixed.

Line 426: "resulting significant" should be "resulting in significant".

Fixed.

Line 480: "11i" should be "11k".

Fixed.

Line 515: "12e" should be "12f".

Fixed

Line 541: Subject verb agreement "loss . . . are".

Fixed to "loss ... is".

Line 578: Typo, "inthe" should be "in the".

Fixed.

Line 582: Typo, "asses" should be "assess".

Fixed.

Line 606: Typo, "arebased" should be "are based".

Fixed.

Appendix B: I think units need to be added to the table.

Units have been added to table caption.

Figure B1 caption: "it's" should be "its".

Fixed.

**References**

Adusumilli, S., Fricker, H. A., Medley, B., Padman, L., and Siegfried, M. R.: Interannual variations in meltwater input to the Southern Ocean from Antarctic ice shelves, Nature Geoscience, 13, 616–620, https://doi.org/10.1038/s41561-020-0616-z, publisher: Nature Publishing Group, 2020.

CICE-Consortium: update_ocn_f problem · Issue #390 · CICE-Consortium/Icepack, `https://github.com/CICE-Consortium/Icepack/issues/390`, 2022.

Naughten, K. A., Meissner, K. J., Galton-Fenzi, B. K., England, M. H., Timmermann, R., Hellmer, H. H., Hattermann, T., and Debernard, J. B.: Intercomparison of Antarctic ice-shelf, ocean, and sea-ice interactions simulated by MetROMS-iceshelf and FESOM 1.4, Geoscientific Model Development, 11, 1257–1292, https://doi.org/10.5194/gmd-11-1257-2018, publisher: Copernicus GmbH, 2018.

Roberts, A. F., Hunke, E. C., Allard, R., Bailey, D. A., Craig, A. P., Lemieux, J.-F., and Turner, M. D.: Quality control for community-based sea-ice model development, Philosophical Transactions of the Royal Society A: Mathematical, Physical and Engineering Sciences, 376, 20170 344, https://doi.org/10.1098/rsta.2017.0344, publisher: Royal Society, 2018.

Susheel Adusumilli, Fricker, H. A., Medley, B. C., Padman, L., and Siegfried, M. R.: Data from: Inter-annual variations in meltwater input to the Southern Ocean from Antarctic ice shelves. UC San Diego Library Digital Collections., https://doi.org/https://doi.org/10.6075/J04Q7SHT, 2020.